

**Extreme droughts and human responses to them: the Czech Lands in the pre-instrumental period**

Rudolf Brázdil[1,2], Petr Dobrovolný[1,2], Miroslav Trnka[2,3], Ladislava Řezníčková[1,2], Lukáš Dolák[1,2],
Oldřich Kotyza[4]

[1]Institute of Geography, Masaryk University, Brno, Czech Republic
[2]Global Change Research Institute, Czech Academy of Sciences, Brno, Czech Republic
[3]Department of Agrosystems and Bioclimatology, Mendel University in Brno, Brno, Czech Republic
[4]Regional Museum, Litoměřice, Czech Republic

*Correspondence to*: Rudolf Brázdil (brazdil@sci.muni.cz)

**Abstract.** The Czech Lands are particularly rich in documentary sources that help elucidate droughts in
the pre-instrumental period (12th–18th centuries), together with descriptions of human responses to
them. Although droughts appear less frequently before AD 1501, the documentary evidence has
enabled the creation of series of seasonal and summer half-year drought indices (SPI, SPEI and Z-
index) for the Czech Lands for the 1501–2017 period. Based on calculation of return period for series
of drought indices, extreme droughts were selected for inclusion herein if all three indices indicated a
return period of ≥20 years. For further analysis, only those from the pre-instrumental period (before
AD 1804) were used. The extreme droughts selected are characterised by significantly lower values of
drought indices, higher temperatures and lower precipitation totals compared to other years. The sea-
level pressure patterns typically associated with extreme droughts include significantly higher pressure
over Europe and significantly lower pressure over parts of the Atlantic Ocean. Extreme droughts with a
return period ≥50 years are described in detail on the basis of Czech documentary evidence. A number
of selected extreme droughts are reflected in other central European reconstructions derived from
documentary data or tree-rings. Impacts on social life and responses to extreme droughts are
summarised; analysis of fluctuations in grain prices with respect to drought receives particular
attention. Finally, extreme droughts from the pre-instrumental and instrumental periods are discussed.

**1 Introduction**
Droughts and floods constitute two extreme aspects of the water cycle. However, while floods are
typified by sudden onset, loss of human lives and immediate material damage, the onset of droughts is
much slower, without direct loss of human lives and result more a chronologically extended range of
impacts, especially on agriculture (agricultural drought), water resources (hydrological and
underground water droughts), and usually with more delay in their broader socio-economic
consequences (socio-economic droughts). The origin of droughts lies in deficit of precipitation totals
compared to climatological norms in a given area (meteorological drought), but it must be noted that
this may exacerbated by other meteorological factors, even by anthropogenic activities (Van Loon et
al., 2016).
        Several extreme drought events with significant human impacts and consequences are known
worldwide from the more recent instrumental period occurring, for example, in Europe in 2003 (Fink et
al., 2004), in Russia in 2010 (Kogan and Guo, 2016), in the U.S. Great Plains in 2012 (Hoerling et al.,
2014; Kogan and Guo, 2016) and in Kenya in 2016–2017 (Uhe et al., 2018). Recent global warming,
arising out intensification of the greenhouse effect due to anthropogenically-enhanced concentrations
of greenhouse gases, may well have contributed to an increase in the frequency and severity of drought
episodes (Dai, 2013). For example, Naumann et al. (2018), in an analysis of drought conditions
corresponding to global warming of 1.5°, 2° and 3°C compared to pre-industrial times, recognised a



progressive and significant increase in drought frequency, particularly in the Mediterranean, most of Africa, western and southern Asia, Central America and Oceania; it is projected that droughts will occur in these regions five or even ten times more frequently than at present.

In Europe, the Mediterranean, a consistently drought-prone region, is considered one of the most endangered areas whether viewed on the basis of observed or proxy data (Cook et al., 2016), or on projections for the 21st century (e.g. García-Ruiz et al., 2011; Seager et al., 2014). However, indications from other parts of Europe are that drought may become a serious hydrometeorological extreme on a far wider scale (Spinoni et al., 2018). This also holds true for central Europe where, quite apart from such outstanding drought events as occurred, for example, in 1947 (Brázdil et al., 2016b),

several serious dry episodes have been recorded in the past two decades (see e.g. Brázdil et al., 2013; Brázdil and Trnka, 2015; Spinoni et al., 2015; Zahradníček et al., 2015; Hoy et al., 2017; Laaha et al., 2017).

       The question remains as to how exceptional are the droughts that occurred in the instrumental period in the context of past centuries or millennia. The frequencies and duration of major droughts

now rank highly among the priorities of modern drought research (Trnka et al., 2018). Among events in the pre-instrumental period, particularly long and extensive extreme droughts have been derived from tree-ring data (e.g., Stahle et al., 2007; Cook et al., 2010a, 2010b), for which the term "megadroughts" has been coined. A degree of controversy with dendroclimatologists has arisen around the extension of this term to a drought that occurred in western and central Europe in 1540 (Wetter et al., 2014)

established by rich documentary evidence (see Büntgen et al., 2015; Pfister et al., 2015). Historical climatology relies largely on such documentary evidence of drought and related phenomena (Brázdil et al., 2005, 2010), in which it has been applied to the study of long-term spatio-temporal drought variability (e.g. Mendoza et al., 2007; Diodato and Bellocchi, 2011; Brázdil et al., 2013, 2016a; Ge et al., 2016; Oliva et al., 2018) as well as for selection and description of various important individual

droughts in central Europe (e.g. Munzar, 2004; Brázdil et al., 2013; Wetter at al., 2014; Brázdil and Trnka, 2015; Kiss and Nikolić, 2015; Roggenkamp and Herget, 2015; Munzar and Ondráček, 2016; Kiss, 2017; Pfister, 2018) and in the other areas of the world (e.g. Dodds et al., 2009; Hao et al., 2010; Zhang and Liang, 2010).

       Although a number of studies of droughts based on documentary evidence already exist in the

Czech Lands (the recent Czech Republic) (Munzar, 2004; Brázdil et al., 2013; Brázdil and Trnka, 2015; Munzar and Ondráček, 2016), the current investigation concentrates on the systematic study of extreme droughts in the pre-instrumental period from the 12th to the 18th centuries. However, the somewhat episodic character of drought information before AD 1500 dictates that the primary focus is confined to extreme droughts during the 16th to the 18th centuries. Sect. 2 of this paper presents

documentary evidence of droughts, long-term series of drought indices and other data sources. Sect. 3 describes the procedure for selection of extreme droughts. The results that appear in Sect. 4 concentrate on selected extreme droughts, the features typical of them, and their detailed description. Sect. 5 discusses the central European context of extreme droughts, their human impacts and responses, and what is to be learned from the extreme droughts of the past. Finally, Sect. 6 summarises the basic

results.

## 2 Data
### 2.1 Documentary data on droughts

A variety of documentary sources may be used for identification of droughts in the pre-instrumental

period in the Czech Lands, i.e. before AD 1804, the year that marks the beginning of mean monthly precipitation series calculated for the territory of the Czech Lands (Brázdil et al., 2012). This development permits the compilation of series of drought indices for the 1804–2017 period, since mean Czech monthly temperature series had already been available from AD 1800 onwards (ibid.).



Information related to droughts in Czech documentary sources may be found in annals, chronicles, memoirs and diaries, weather diaries, financial-administrative records, religious sources, songs, newspapers and journals, society reports, epigraphic evidence, chronograms and early instrumental measurements (see e.g. Brázdil et al., 2013; Brázdil and Trnka, 2015).

Information concerning the beginning, course and end of drought episodes is usually relatively brief in documentary evidence. This is particularly true of reports in narrative sources. For example a chronicle kept by Pavel Mikšovice from Louny (for places reported in this article see Fig. 1) says of the 1540 drought (AS17 – archival source AS17): "*In that year* [1540], *drought and heat were so severe that there was no rain during the entire summer from Holy Ghost* [16 May] *even until the Thursday*

*after Saint Jacob* [29 July] *and then, again, no rain occurred for a long time, right up to the Tuesday after Saint Francis* [5 October] *and high prices followed on from this* […]" More detailed information can be obtained only from weather diaries (e.g. AS4 and AS7) in which daily weather records appear, or from early instrumental meteorological measurements (Brázdil et al., 2012).

On the other hand, records related to drought impacts are more frequent and detailed in

documentary evidence. They usually reflect the lack of water and associated difficulties as well as problems with harvests. For example, according to an ancient "book of memory" from Litoměřice (Smetana, 1978), the summer of 1503 was so dry "*that people could not remember such* [a] *dry* [summer] *for 30 years, since they could not mill on many brooks and rivers and there was a bad harvest in the fields; there was almost no spring grain because, in many places, the grain had to be*

*plucked as it was impossible to reap it. And the wine was very good that year* […]"A town scribe in Litoměřice reported a lack of water in autumn 1548 (Smetana, 1978): "*This year there was so little water in the Elbe that nobody could recall it* [lower]. *At Roudnice* [nad Labem] *the water stood* [still] *above the weir on the Friday of St. Andrew's* [30 November, i.e. 10 December of the Gregorian calendar] *from morning to as late as afternoon. And people below the weir seized and caught fishes by*

*hand.*" Low water on rivers put water-mills out of operation, while shortages of drinking water occurred, as follows from an entry for 1746 in the book of memory kept by the Chládek family for Nové Město na Moravě (Trnka, 1912): "*There was such a drought the whole year* [1746] *that it did not rain the whole summer from spring* [onwards]*; also, there was no harvest of spring cereals except rye, it was impossible to mill anywhere*, [and grain] *was transported as far as 6 or 7 miles* [i.e. 45 and 52

km respectively] *to be milled,* [and] *because all the wells dried up, neither people nor livestock had anything to drink.*" Droughts were often accompanied by forest fires as reported by František Václav Felíř, a Prague burgher, for 1746 (Vogeltanz and Ohlídal, 2011): "[…] *It has not rained since the first of June to this day* [end of July]*, only twice so little that it could only wet the dust, and so overwhelming was the sweltering weather, that through great desiccation and drought forests caught*

*fire of their own accord, as did those of Prince Mansfeld.*"

The impacts of severe droughts extended beyond immediate agricultural concerns into the realms of urban society and personal matters. A striking example of social-cultural response to drought is documented by a report from Prague dating to 15 July 1503 that appears in *Staré letopisy české* [the "Old Czech Annals"] (Palacký, 1941): "[…] *priests and Utraquist noblemen in Prague declared a fast*

*in order that Lord God might condescend to send down rain. But those who side with Rome desired neither to keep a fast nor render the day holy, and many* [of them] *who were in Prague went to* [Prague] *castle, and some to Lesser Town* [Malá Strana]*, to eat meat. And God sent no rain because* [people] *had prayed to him without concord and unity.*"

## 45  2.2 Series of drought indices

A variety of drought indices are used to describe drought patterns (see e.g. Brázdil and Trnka, 2015; Svoboda and Fuchs, 2018). Among the most frequently used are the Standardised Precipitation Index SPI (McKee et al., 1993), the Standardised Precipitation Evapotranspiration Index SPEI (Vicente-



Serrano et al., 2010), the Z-index and the Palmer Drought Severity Index PDSI (Palmer, 1965), usually calculated from measured precipitation totals and temperatures. Seasonal (winter – DJF, spring – MAM, summer – JJA, autumn – SON), summer half-year (April–September) and annual series of these four drought indices have been calculated for the territory of the Czech Lands for the years after AD 1501 by Brázdil et al. (2016a). Their calculation applied monthly temperature series for central Europe by Dobrovolný et al. (2010) and seasonal precipitation series for the Czech Lands by Dobrovolný et al. (2015), both based on reconstruction of temperature/precipitation indices series derived from documentary data and instrumental measurements. More recently, these drought indices series were extended up to March 2018.

**2.3 Other comparative series and datasets**
To compare selected extreme Czech droughts from the pre-instrumental period with other related Czech and central European data, the following series were used (always from AD 1501 onwards):
(i) JJA scPDSI derived from tree-rings and included in the European Old World Drought Atlas – OWDA (Cook et al., 2015), from which gridded data were used to calculate series for the Czech Republic (91 grids) and central Europe (421 grids)
(ii) March–July precipitation totals reconstructed from tree-ring series of fir (*Abies alba* Mill.) from South Moravia (Brázdil et al., 2002 – extended)
(iii) May–July precipitation totals reconstructed from tree-rings of oak (*Quercus* spp.) in Bohemia (Dobrovolný et al., 2018)
(iv) April–August SPEI series reconstructed from grape harvest dates in the Bohemian wine-growing region by Možný et al. (2016).
Because of the strong influence of droughts on agricultural production, low harvests or crop failure may be reflected in grain prices. For comparative purposes, the following series of grain prices were used:
(i) series for wheat and rye in certain Moravian towns during the 1540–1622 period (Novotný, 1963)
(ii) series for wheat, rye, barley and oat prices for the royal town of Dačice in the 1625–1802 period (Brázdil and Durďáková, 2000)
(iii) series for wheat, rye and barley prices for the 1655–1872 period in the city of Prague (Schebek, 1873).

**3 Methods**
Reconstructed series of drought indices from AD 1501 onwards (see Brázdil et al., 2016a) and extended up to 2018 were used in the selection of extreme droughts, based on calculations of return periods (re-occurrence intervals). The advantages of employing the above drought indices include clear information about drought intensity, the opportunity to compare pre-instrumental and instrumental dry episodes at the same level, and the way in which droughts from the pre-instrumental and the instrumental periods may be considered in parallel. On the other hand, the onsets and terminations of drought episodes beyond the standard seasonal or summer half-year limits may be taken as a certain disadvantage.
Calculation of N-year return period for MAM, JJA, SON and summer half-year (April–September) drought indices series was based on the peak-over-threshold (POT) approach. Since this study addresses extremely low values (minima representing the driest years), the first step was to transform negatively (multiply by −1) the drought index series. Values above the 80th percentile were then used as the samples for further analysis. As follows from Extreme Value Theory, high values above a sufficiently high threshold may be reliably modelled with the Generalized Pareto Distribution (Coles, 2001). Consecutive parameters of the distribution were estimated and their suitability, together with the appropriateness of the threshold value according to various diagnostic graphs, were tested with





the in2extRemes package (Gilleland and Katz, 2016). Finally, return values of the individual drought indices for return periods of N = 10, 20, 50, 100, and 200 years were calculated and transformed back to minimum extremes (Fig. 2).

In order to select extreme droughts, only those corresponding to at least a 20-year return period according to all three drought indices (SPI, SPEI, Z-index) were taken into account. This basic dataset of extreme droughts was also supplemented by some cases in which the above condition was fulfilled for only SPEI and Z-index, but concurrently an SPI of at least N = 10 years return period was achieved (see Table 1). For the study of the features typical of extreme droughts in Sect. 4.1.2, these were analysed together with respect to their severity (after an N-year return period), meteorological

features (box-plots of drought indices, temperatures and precipitation) and synoptic patterns (maps of sea-level pressure patterns in the Atlantic-European area) based on Luterbacher et al. (2002); data covering the 1500–1999 period are available at https://www.ncdc.noaa.gov/paleo-search/study/6366, last access 20 September 2018). Various Czech documentary data were further used to describe the most outstanding summer half-year and seasonal droughts (N ≥50 years) separately, based on more

restricted evidence (Sect. 4.2.2); full existing evidence of these events is far too extensive to be included within the scope of the current contribution.

## 4 Results
### 4.1 Extreme droughts in the Czech Lands during the pre-instrumental period
#### 4.1.1 Droughts before AD 1500
Only 36 drought episodes before AD 1500 were identified for the Czech Lands (particularly in Bohemia). The first historically credible drought report from the Czech Lands occurred in DJF 1090/1091, for which Cosmas, the canon of the Prague chapter, reported no rain and no snow for that winter (Bretholz, 1923). With the exception of this first record, all further drought entries are related to

MAM, JJA or to the whole summer half-year, in relation to the negative impacts of droughts, particularly on grain harvests (see Table 1 in Brázdil et al., 2013). The written reports of these droughts usually lack the detail necessary to enable comparison of the events with respect to their onset and duration, course, severity or impacts. Their total number is considerably underestimated due to scarce documentary data (see Brázdil and Kotyza, 1995) and it is difficult to identify cases that could clearly

be classified as extreme.

#### 4.1.2 Droughts in AD 1501–1803
Seasonal (except DJF) and summer half-year series of drought indices were used to select extreme droughts according to calculated N-year return period (see Sect. 3). From the overview of selected

extreme droughts in Table 1 it follows that use of different types of drought indices can generate different return periods, as priorities vary between higher dependence on precipitation regime (SPI) or combined precipitation-temperature effect (SPEI), or also taking soil characteristics (Z-index) into account. Extreme droughts exhibit quite contrasting seasonal distributions over the centuries: half of them occurred in the 18th century for MAM, and a quarter each in the two remaining centuries; for

JJA, the maximum occurred in the 16th century (44%) and minimum (25%) in the 18th century. The SON extreme droughts are distributed almost identically over the three centuries. The 200-year droughts were most frequent in MAM, with 11 cases (but 100-year drought only once), while for the other two indices the corresponding frequencies were 6 vs. 3 for JJA and 4 vs. 7 for SON. Droughts of N ≥100 years according all three indices were recorded only for 1638 and 1779 in MAM, 1540 and

1590 in JJA, 1727 in SON, and 1540, 1590 and 1616 in the summer half-year.

Fig. 3 provides composite information in the form of box-plots for selected groups of seasonal extreme droughts with respect to the dataset for all the remaining years. For example, box plots for MAM are created from 16 selected years from Table 1 on the one hand and all 287 remaining years of



the 1501–1803 period on the other. Box-plots created for SPI, SPEI and Z-index are further supplemented by seasonal mean temperatures for central Europe (Dobrovolný et al., 2010) and seasonal mean precipitation totals for the Czech Lands (Dobrovolný et al., 2015). Lower values of drought indices, higher temperatures and lower precipitation in the extreme droughts group compared with the remainder of the datasets are typical for all three seasons (MAM, JJA and SON). Differences in mean seasonal drought indices, temperatures and precipitation in the drought groups are statistically significant at the 0.05 significance level (t-test).

Sea-level pressure (SLP) data by Luterbacher et al. (2002) were used to elucidate synoptic patterns for cases of extreme drought. For every season, a composite of mean SLP was created (Fig. 4, part a), only for selected years with extreme droughts. Mean SLPs for 1961–1990 were constructed for reference values (Fig. 4, part b) and differences between the two types of map were further calculated (Fig. 4, part c) and tested with respect to their statistical significance ($\alpha = 0.05$). The mean SLP field for an extremely dry MAM shows a broad ridge of high pressure extending north-easterly and easterly from the Azores High over the European lands. A similar ridge of high pressure is also typical of JJA, but the pressure decreases from west to east. The mean SLPs for SON extreme droughts are characterised by a broad belt of high pressure over Europe in which an isolated anticyclone appears in south-eastern Europe. An increase in SLP over the European continent (with the exception of northern Europe) is clearly expressed for extreme droughts in comparison with the reference when differences are statistically significant. Concurrently, statistically significant pressure decreases appear in the Atlantic Ocean.

**4.2 Descriptions of selected extreme droughts**

Only extreme droughts with return periods of at least ≥50 years for one of the three indices used were selected for more detailed description of patterns. The descriptions start from the summer half-year; this means that extreme seasons included in any summer half-year are not described again in any following seasonal account. For example, the year 1540 appears as outstanding in MAM, JJA and SON but it is not described for every season separately. The descriptions themselves usually cite only sufficient examples of reports to characterise events, since including all the Czech documentary information available for each event lies well beyond the possible scope of this article. If related archival sources have already been published, they have been preferred in quotations before the report of the original archive location.

**4.2.1 Extreme droughts of the summer half-year**

**(i) 1536**

According to a source in Litoměřice, the summer of 1536 was very dry and the water in the River Elbe ran low. Considerable damage was done by an outbreak of caterpillars. Roses blossomed twice (Smetana, 1978). Pankraz Engelhart, the chronicler for Cheb, reported summer drought and heat with frequent fires in forests and settlements, and also a dearth of grain after a poor harvest, although ample quantities of good wine, fruits and nuts were available (Gradl, 1884). Marek Bydžovský of Florentinum (Kolár, 1987) mentioned dry weather in Bohemia from April to Christmas, with wells and streams drying up and frequent wildfires. In his records for south-eastern Moravia, Lord Jan of Kunovice reported warm weather and great drought lasting until 24 November (Brázdil and Kotyza, 1996).

**(ii) 1540**

A long period of warm, dry weather is reported by many documentary sources for 1540. The daily records kept by Jan of Kunovice indicate a very dry spring, recording only four days with snowfall in March, four rainy days in April and no rain at all in May, with several entries for "drought" (Brázdil and Kotyza, 1996). A plethora of documentary sources highlight a hot, dry summer, shortages of water,





early harvests and frequent forest fires. Martin Leupold von Löwenthal, the town scribe in Jihlava, reported a dry period from 6 April until the end of the year, with bad yields of vegetables and beet (d'Elvert, 1861). Very dry and warm conditions with severe shortages of grain and vegetables occupied much the same time period in Uherský Brod (Zemek, 2004). Pavel Mikšovice (AS17) reported a hot,

dry period in Louny from 26 May to 13 October with rain on only 8 August (see report in Sect. 2.1). A poor or medium harvest of grain was also mentioned (Gradl, 1884; Kolár, 1987). On the other hand, it was an excellent year for wine in Bohemia, with an abundant harvest of grapes (AS16).

**(iii) 1590**

The extreme drought of 1590 was described in the "book of memory" of Litoměřice as follows (AS16):

"*In that year there was great drought from Holy Ghost* [10 June] *right to* [the festival of] *St. Matthew, Apostle of God* [21 September]; *thus it rained hardly twice or thrice, the* [River] *Elbe fell and was so small, that the Elbe could be walked over or crossed by wagons, by horse and on foot. The water was spoiled and green, so people could not use it with any comfort, since to some green, to others yellow, seemed unhealthy* […]" An entry from Soběslav records the negative impacts of the 1590 drought on

agricultural products (AS12): "*Such was the drought that all the earth became parched, the grass scorched sere, spring and autumn grain parched and dried up, and garden crops could not grow either* […]. *Then in this year great and unheard-of famine developed.*" Periods of great heat and drought were also accompanied by forest fires, as recorded by Pavel Mikšovic from Louny (AS17): "*In that year great heat made the forests catch fire at Království* [near Louny]; *the lord of Weitmille lost many*

*forests worth many hundred* [gulden] *due to fires* [caused by the] *heat in the Meissen mountains* [Krušné hory Mts.]."

**(iv) 1616**

The chronicle of the Podolský family from Drahotuše (Indra and Turek, 1946) describes the drought of 1616 as follows: "*There was a great drought that year, starting straight after Holy Ghost* [22 May]

*until nearly Christmas, out of which great shortages for mills as well as of foodstuffs arose; of the grain there was nearly nothing.*" Martin Leupold von Löwenthal reported a very dry summer with shortages of water in Jihlava, where the fish-cultivation ponds ran water off to provide water for mills (d'Elvert, 1861). It was very dry from 19 April until Christmas in Fulnek; lack of water for water-mills forced people to travel great distances to process what grain they had (AS10). The weather was dry for

the whole year after spring in Rožnov pod Radhoštěm (AS9). Similarly, Pavel Mikšovic (AS17) reported a period without precipitation between 3 April and 31 July in Louny. Daniel Basilius of Deutschenperk, a university professor, mentioned great heat, dried-up rivers and the River Vltava "stinking" at Prague (Winter, 1899). The Reverend Daniel Philomates the Elder spoke in a sermon (Fig. 5) of a 100-year drought. The year 1616 is clearly indicated by a mark on the hunger stone in the

River Elbe at Děčín that indicates the low water level (Fig. 6).

**(v) 1631**

The chronicle kept by Pavel Mikšovic in Louny includes a short description of the 1631 drought (AS17): "*That year, around the time of Saints Peter and Paul* [29 June], *before and after,* [such] *a tremendously great drought occurred* [that] *in some places quite large streams, as well as springs,*

*dried out; in many villages, not having water, people had to go some distance for water for their livestock. It was impossible to mill anything on many streams; in many places forests caught fire and burned due to great heat.*" Michel Stüeler (Brázdil et al., 2004; Kilián, 2013), a master tanner in Krupka, wrote of a drought, and for the most part dried-out spring grain, in a summary report for his memorial book for 1631. Meadows became dry and there was no fodder (ibid.). Of a fire in Načeradec

on 20 May, local scribe Václav Smrž remarked "*that due to great drought nothing remained*" (Teplý, 1928b).

**(vi) 1727**





That the summer half-year of 1727 exhibited dry patterns is evident from the daily records kept by the Premonstratensian monks in the Hradisko monastery in Olomouc (AS7). Drought was already apparent in April; the entry for the 26th included: "*much-needed rain*". The following month had periods of extreme heat, so "terrible" that the entry for 14 May relates weather so dry that the earth cracked. After

further episodes of heat in June and July, persistent clear, hot weather continued from mid-August to the end of September, when the phrase "*much-needed rain*" reappeared in the record for 4 September. Very low levels of water in various parts of Moravia, mentioned for 12 September, gave further indication of the extended hot and dry period. This is confirmed by the very low frequency of recorded days with precipitation: six in July, four in August and three in September (see Brázdil et al., 2011).

According to municipality financial records, two masses for rain were held in Vřesovice (Opletal, 1933). Records from Frenštát pod Radhoštěm (Strnadel, 1950) report great dearth, hunger and very hard times in 1727, noting that the winter cereals perished in heavy snow in the winter and the spring cereals were killed off by drought in summer throughout Moravia, as well as in nearby provinces.

**(vii) 1728**

Entries in the Premonstratensian diary kept by the Hradisko monastery at Olomouc for 1728 (AS7; Brázdil et al., 2011) indicate similar drought patterns as in the previous year. As well as reports of persistently very hot weather during the summer months, only four rainy days were recorded for June, same as for July, while in August it rained on only three days (ibid.). A secondary source (Noháč, 1911) reports drought in 1728, together with the previous year, for the Břeclav region. Moreover, in

1728 "*hordes of locusts* [appeared] *in the fields, which devastated the scanty remnants of the yield*" (ibid.).

**4.2.2 Extreme seasonal droughts**

**4.2.2.1 Spring droughts**

**(i) 1571**

Dry months for April and May follow from the records of Jan Strialius, a scribe in České Budějovice (Brázdil and Kotyza, 1999). In addition to noting that April was "*as sunny as May*" he indicated drought in a record for 30 April, great drought for 8 May and on 3 June ("*thus far, great drought*"). Such dry weather may also be confirmed by the writing of the knight Pavel Korka of Korkyně (Vybíral,

2014), who reported bad yields for Bohemia (particularly wheat and rye), a lack of hay, and shortages, but a good yield of grapes (see also Brázdil and Kotyza, 2014). Shortages and hunger were also reported in many other Bohemian and Moravian narrative sources, but without specific details.

**(ii) 1603**

According to records kept by Jan Voldřich Klusák from Radovesnice (AS8) "[…] *great drought*

*immediately after Easter* [30 March] *right until the grain harvest, such that older people maintained that they could remember none greater, since grasslands, and also meadows in some places, burned and it was possible to harvest only scant straw and hay* […]" Almost no rain from spring to the grain harvest, when the winter cereals existed only upon winter moisture and grain was therefore expensive, were reported in the chronicles of an Anabaptist sect, the "Hutterian Brethren", in Moravia (Wolkan,

1923). This is confirmed by a report of a dry year from the chronicle of the Podolský family at Drahotuše (Indra and Turek, 1946) that "*nearly nothing came of the wheat and oats because everything remained near the ground.*"

**(iii) 1638**

Spring 1638 was identified by all MAM drought indices as a very extreme (200-year) drought. Its

indicative patterns are made clear by a record noted by Václav Nosidlo of Geblice (Lisá, 2014): "*Great periods of heat and drought began on the 7th day of April, not dissimilar from those that sometimes*



*appear in the dog days* [i.e. 14 July–15 August]*, thus to this day* [7 June] *a large part of the spring crops, particularly in Bohemia, has withered; the winter cereals have ripened without grain. When in the course of these days the Lord provided moisture, it became colder and morning frost followed [...] At these frequently-sprinkled times, the air was fresh* [and also] *the cereal that stood already dry and*

5       *looked as if it had no grain improved again greatly and became better, which was surprising.*" Great drought leading to grain and grass drying out in many places is described in letters sent by Václav Králík, administrator of the Nové Město nad Metují domain (Šůla, 1998): on 17 May he says "*it has not rained for five weeks now*", making further note of dryness in the letter from 28 May. A letter from Martin Škvorecký, an administrator from Pacov, to Lady Zuzana Černínová, dated 16 May (Teplý,

1928a), describes matters in similar terms: "*God's* [harvest] *of winter rye and wheat becoming burned due to extremely hot and dry weather, spring grain similarly. If this continues* [any] *longer, everything in the fields will mature without profit. The grass also appears bad and cannot grow due to great drought.*" The annals of Jan Čeledínek from Čáslav (AS1) report such great drought that it did not rain from 11 April until the end of June. Forest fires broke out due to great drought in the surroundings of

Křivoklát (Nožička, 1957). The chronicle from Holešov (Fialová, 1967) reports "[such] *an enormously great drought that in many places wheat could not grow and come into ear, and it remained in the form that it grew in spring; then the dearth started.*"

**(iv) 1683**

According to a "Councillor Manual" from Křinec, there was no rain from spring onwards and harvest

failure threatened. This was serious enough to organise a procession of entreaty to the Holy Trinity for rain. A meeting of the municipal council on 11 June deputised a number of people to arrange finance for it. Heavy rain fell a day later (Hellich, 1905).

**(v) 1686**

According to records of the Jesuit college in Klatovy (Peters, 1946), the weather was dry for some

months. The chronicle or Rýmařov (AS11) reported a grain failure due to extended drought. Low water in the rivers prevented rafts from transporting grain and flour to Prague (Holec, 1971). According to the chronicle of Mikuláš František Kernerius from Hnojice (Prucek, 1985), dry weather had already done great damage to the grain before harvest time (7–8 July) when, although some rain fell, it wetted the soil only a little. However, when on the few days showers fell on already-cut shocks of wheat and rye,

some grain started to show through (ibid.).

**(vi) 1779**

The memoirs of the Brodský family from Roudnice nad Labem report no snow and rain for February–April, warm weather in March–April and rye with short stalks due to drought (Kopička and Kotyza, 2009). A report by František Tomáš Spillar, a teacher, in a memorial book from the Plzeň area reads as

follows (AS13): "*Immediately after Candlemas* [2 February] *it was warm and* [there was] *drought without rain, only once did it rain in spring,* [and the] *grain in the fields dried. Then, as late as on the 10th of the June month, it rained, before which* [people] *from nearly all the parish churches had held processions to various places to obtain moisture.*" In Javorník, it did not rain from 2 February to 24 April then, after slight precipitation, the drought continued until a rainy period started on 25 June

(Paměti starých písmáků moravských, 1916). Florián Velebil reported a great drought, with no rain from 30 April to 16 June in Městec Králové (Robek, 1978). The parish records for Bruzovice mention a great drought from February to the end of May, followed by rain and floods in Silesia (Pospíšil, 1905). According to the memoirs of Karel Josef Voda from Hlinsko, the spring cereals could not even emerge from the ground due to terrible drought in spring, and what eventually grew in some places had to be

plucked rather than reaped (Adámek, 1917). The records of Jan Nepomuk Hausperský from Brno report nice dry weather in February, unusually lovely days in March and April with fully blossoming trees in gardens and forests, and finally warm and dry weather in May (Brázdil and Valášek, 2003). According to the newspaper *Brünner Zeitung* (1779, No. 44, p. 348) of 3 June, an unremitting drought



reduced the water level in the River Vltava to the point at which transit across the river was rendered impossible by the height of the banks above the water in many places. A large number of mills could not operate and various fish-cultivation ponds dried out and the fish perished (ibid.).

**(vii) 1794**

Anton Lehmann, a teacher from Noviny pod Ralskem (AS15) reported that in April 1794 the days were as hot as in June and that the dry soil needed rain, which remained absent until 9 May. A plague of caterpillars consumed nearly all the leaves from the fruit trees. Dry, hot weather led to only small yields from summer seeds and garden fruits. The aftermath also suffered badly (ibid.). The memoirs of Karel Josef Voda from Hlinsko (Adámek, 1917) say of 1794 that "*terrible drought and heat,* [such]

*that there was no rain from the month of May up to the time of harvest in July; none of the grain grew well, the flax was affected sorely and then burned out* [...]" In reference to a catastrophic fire on 27 April in Bystřice nad Pernštejnem, the local chronicle reported great drought, saying it not rained for more than two months (Paměti starých písmáků moravských, 1916). Spring drought is also mentioned at Rožnov pod Radhoštěm, where there was no rain for over three months (AS9).

**(viii) 1800**

Josef Schück (Bachmann, 1911) reports great drought in Litoměřice, recording that from Christmas (1799) to September did not rain more than twice; the River Elbe dried out and the hunger stone at Žalhostice (installed in response to the 1660 drought) appeared. A period without rain from Christmas to 16 May was reported by Florián Velebil for Městec Králové (Robek, 1978). Anton Lehmann from

Noviny pod Ralskem mentioned a very dry spring with a bad harvest of summer crops (AS15). The chronicle from Nové Město na Moravě (Trnka, 1912) mentions a warm, dry May, but dryness and cold for June and July. As a result of severe drought, milling was a problem all year. Although the harvest of rye was good, the flax dried up and burned out. This source also reported dry weather in Austria and Moravia leading to a very bad harvest (ibid.). The chronicle of Jan Čupík from Olešnice (AS14) notes a

dry, but good, year. The "book of memory" kept by the Augustinians of Klatovy (Řehák, 1912) mentioned a procession of entreaty for rain from Domažlice on 25 August: when it returned, rain started. This source also mentions drought for the whole of Europe so severe that many mills did not operate due to lack of water (ibid.). A very low water-level on the River Elbe is marked on the hunger stone at Děčín (Fig. 6).

**4.2.2.2 Summer droughts**

**(i) 1630**

A summary record written by Michel Stüeler in the memorial book of Krupka (Kilián, 2013) indicates summer and autumn 1630 so dry that all the streams and springs ran dry. He even maintains that this

drought was greater than that of summer 1590. Despite the drought, the harvest of winter grain was good, and average for spring grains. There was an excess of grapes and sufficient fruits, nuts and further crops (ibid.). Compared to Krupka's records, the chronicle kept by Pavel Mikšovic for Louny reported the average harvest of winter grains and due to a great drought a small harvest of field crops (AS17).

**(ii) 1684**

The chronicle of Michael Heger, the weaver in Moravská Třebová, reported such a hot, dry summer for 1684 that "*summer crops perished, particularly barley, oats and flax*" (Spina, 1905). According to Jesuit records, drought reduced yields of crops in many places in the Kutná Hora region (Podlaha, 1912).

**(iii) 1746**

The records of František Václav Felíř from Prague (Vogeltanz and Ohlídal, 2011) report only two episodes of rain between 1 June and the end of July, with hot, dry weather prevailing otherwise (see his report in Sect. 2.1). A procession of entreaty for rain was organised on 9 July. The grain dried up and





frequent forest fires broke out. Enough rain to be "useful" fell as early as 18 August, but dry weather then set in again (ibid.). A drought almost beyond living memory was reported in Litoměřice. Streams dried up and the River Elbe fell to unusually low levels; lack of water for mills then led to shortages of flour and bread. The grain harvest was bad, as was that of fruits. Many trees dried out (Katzerowsky,

1887). The lack of water in the Elbe is indicated by a mark on the hunger stone at Děčín (Fig. 6). Dry patterns, bad harvests, lack of water and problems with milling were reported from Nové Město na Moravě (see Sect. 2.1). Hieronymus Haura (AS6), a member of the Augustinian order in Brno, mentioned a devastating drought and great heat in June and particularly in July, as a result of which people died. Grass and grain hardly grew (better for grain than for straw), fruits fell with their leaves,

the earth was cracked, the roads were very dusty, springs and streams dried up, transport on larger rivers was interrupted, and water-mills were out of operation with a consequent lack of bread. Processions of entreaty for rain were organised in Brno on 17 July and 8 August. Haura noted that it was sad to see how summer was recalled by a late autumn without grass, blossoms, leaves or fruits. He created a chronogram that reflected this experience of drought (see Brázdil and Trnka, 2015).

### 4.2.2.3 Autumn droughts

#### (i) 1548

A dry autumn 1548 may be deduced from reports of low water on the River Elbe. Thus a report dated 10 December from Litoměřice mentions a water level so low that the water was "*standing still*"

(Smetana, 1978). At Ústí nad Labem it was possible to cross the bed of the River Elbe "barefoot" before and on 10 December (AS18). This tallies with a report from Jan Jeníšek, a landowner, who mentioned very little water in the fish-cultivation pond near Svrčovec around 8 November, citing severe summer drought as the reason. He noted good fields for 15 November, but drought (AS2). Due to extremely dry conditions there were only few pheasants in the vineyards around Most (Nožička,

25 1962).

#### (ii) 1605

A dry autumn for 1605 is indicated by anonymous daily weather records, probably originating in Prague or nearby. Precipitation days were indicated only on 24 September (otherwise generally warm and sunny weather), on 16 and 27–28 October, and on six days from 14 November onwards. However,

there are no weather entries between 4 and 13 November (AS4).

#### (iii) 1634

A very dry year, but with a cheap grain, was reported by Michel Stüeler in a "book of memory" at Krupka (Kilián, 2013). Mauder (1930) cited 1634 as among the years with low water levels on the River Elbe at Děčín.

#### (iv) 1680

Dean Bartoloměj Zelenka from Soběslav made irregular weather notes (Brázdil and Kotyza, 2001) and mentions "*warm as in mid-summer*" for 12 April. At the same time he indicated that drought had made it impossible to sow. Continuous drought was then reported consistently from April to September. The grain harvest started on 5 July, but drought led to low yields of barley and oats. In August, drought also

hampered ploughing. Although he reported great drought again on 2 October, he mentioned grain growing well three days later (ibid.). In Hnojice, according to a chronicle kept by Mikuláš František Kernerius, a plague of caterpillars until June consumed all the leaves on the fruit trees (Prucek, 1985). Secondary sources report drought and grain failure at Postoloprty before 14 August (Veselý, 1893) and a great drought at Krupka (Bervic and Kocourková, 1978).

#### (v) 1686

Heinrich Teigel, a pharmacist in Litoměřice, noted a nice autumn with a second blossoming of pears trees on 13 October and an average yield of very good wine (Katzerowsky, 1895). The chronicle of Mikuláš František Kernerius from Hnojice (Prucek, 1985) explicitly reported a dry autumn which,



together with previous dry weather, meant that there were nearly no fruits and a bad harvest of the majority of field crops such as grain, beet and cabbage ("*spoiled and faded by drought, many as if scalded*"). An abundance of pests such as mice also contributed. Millers could not mill for lack of water, leading to a lack of bread. For example, a serious lack of water in Bouzov led people, literally

one after the other, so steal from a well. Those who could, transported barrels of water from quite a distant brook (ibid.). The chronicle of Rýmařov also mentioned a hot, dry year with the prices of foodstuffs rising (Anonymous, 1937). Jesuit notes from Klatovy reported a great drought lasting several months (Peters, 1946). Low levels in the rivers made it impossible to transport grain and meal to Prague by water (Holec, 1971).

**5 Discussion**
**5.1 The central European context of extreme droughts**
The selected extreme Czech droughts covered in this paper may be compared with extremes from other drought-related series. Table 2 shows the overall attribution of extreme pre-instrumental droughts to the

first ten and second ten most extreme of them. For example, among the first ten most extreme droughts in the Czech Lands for the summer half-year (AMJJAS) appeared, in order of decreasing drought severity, those of 1540, 1590, 1616, 1947, 1727, 1842, 1536, 1868, 1631 and 1834, of which those of the pre-instrumental period (1501–1803) were included in Table 2. The inclusion and ranking order of a given drought to sets of MAM, JJA and AMJJAS series were based on calculation of the weighted

mean in series of three drought indices (SPI, SPEI and Z-index). Extremes corresponding to the same pre-instrumental period derived from other series were employed for comparison with selected extreme droughts from the current paper. While the AMJJA SPEI series by Možný et al. (2016) are based on documentary data of grape-harvest dates from the Bohemian wine-growing region, the four remaining comparative series are based on tree-ring widths. These include two precipitation reconstructions for

MAMJJ in South Moravia (Brázdil et al., 2002) and for MJJ in Bohemia (Dobrovolný et al., 2018). The other two JJA scPDSI reconstructions were derived from OWDA (Cook et al., 2015), calculated for both the territory of the Czech Republic and central Europe (Brázdil et al., 2016a).

     As might be expected, all the extreme droughts reconstructed for the Bohemian wine-growing region (Možný et al., 2016) are also reflected in selected droughts in the current paper (Table 2). From

13 pre-instrumental droughts based on oak tree-rings in Bohemia (Dobrovolný et al., 2018), agreement with those in the current paper occurs in eight cases. Three remaining tree-ring-based series exhibit agreement with them in a half the cases. The topical discussion of a "megadrought" in 1540 by Büntgen et al. (2015) and Pfister et al. (2015) appears justified, since the event was the most severe in all documentary-based datasets, except in MAM, where 1540 is the third most severe. While the

megadrought does not appear among the 20 most severe cases in two Czech tree-ring series, it corresponds to the 12th order for the territory of the Czech Republic and to the 17th order for central Europe in series derived from OWDA.

     This draws significant attention to the importance of including all available sources in drought reconstructions, as the true frequency and severity of given events may easily be over- or under-

estimated. Pre-instrumental reconstructions, in particular, rely on data sources that are intrinsically of uneven distribution across the studied area. While droughts are in general considered large-scale phenomena, major events may be relatively local and easily missed or exaggerated. One illustrative example may be found in the major drought event of 2011/2012 that strongly affected the eastern Czech Republic (*c.* 1/3 of Czech territory), while the rest of the country remained virtually untouched.

However, impacts in the area afflicted by the drought were particularly severe, with some regions recording cereal yield losses or wildfires unprecedented for half a century (e.g. Zahradníček et al., 2015).



### 5.2 Impacts on social life and human responses to extreme droughts

As follows from Sect. 2.1 and 4.2, extreme droughts have influenced a range of economic and financial activities, as well as impacting on the quality of human lives. Such impacts have not only been caused by reduced precipitation (meteorological drought), but have also been intensified by other

meteorological variables (temperature, evapotranspiration, wind speed) and further modified by the landscape character as well as human activities. Reported below are only those drought impacts and responses that are known from Czech documentary evidence.

The direct impacts of absent or low precipitation were soon reflected in decreasing soil moisture with negative impacts on the growth of crops (agricultural drought). This may have resulted in

complete failure of the crops, or a bad harvest, to be followed by lack of seed for future use. Problems also arose when the availability of straw was limited. Damage to crop yields was sometimes exacerbated by population explosions of small rodents arising out of dry weather. Dried-out pastures, bad hay and poor aftermath harvest meant lack of feed for livestock, so fodder had to bought at inflated prices or it became necessary to decrease the number of stock by selling at below-normal market

prices. Drought also influenced the very nature of agricultural work: sometimes poor grain crops had to be torn off by hand instead of being reaped; soil cultivation and sowing into hard, dry soil was very difficult or even impossible. Such impacts on agriculture lead to rising financial costs and losses to farmers at times when no compensation lay in price increases for agricultural crops and other goods (see e.g. Dolák et al., 2015).

Intense droughts also had negative effects on fruit trees and forests. Fruit trees were often attacked by eruptions of caterpillars and certain other pests, often eating their leaves or invading fruit bodies, rendering them, small, maggoty, and prone to fall prematurely. In forests, the numbers of dry trees increased after drought events, while newly-planted young trees dried out. Moreover, dry episodes are among the factors that contribute to the occurrence of bark-beetle calamities. Dry patterns also

favour frequent and extensive forest fires. Beyond that, if fire that broke out in any construction or store, lack of water made it difficult to extinguish.

Following such a range of direct effects on agriculture, lack of precipitation became apparent in the reduction of essential water resources (hydrological and underground water drought). Drying rivers and/or their low water levels could bring essential water-borne transport to a halt. The operation of

water-mills would cease, with consequent lack of flour and therefore bread. In many cases people had to travel great distances with whatever grain they had salvaged to have it milled. When springs, wells, fountains and other water sources dried out, drinking water for people and animals was simply not available locally; water had to be transported from distant places or scarce capital expended on buying it. Low or standing water, combined with high temperatures, contributed to deteriorating water quality

(green or stinking water) while associated decreases of the oxygen content of fish-cultivation ponds led to extensive fish kills. It should be noted that the "fish-cultivation pond" has been a feature of the Czech economy for many centuries. These carefully-tended, often man-made, bodies of water, sometimes very extensive, have always constituted a major source of first-class protein in a land-locked country.

It follows from the above that the impacts of droughts also had important socio-economic consequences (socio-economic drought), in the form of food shortages, increase of the price of grain, other crops and goods, requests for tax reduction, poverty, debt, distress, and even fully-blown famine. Prohibition of grain exports and on the distillation of spirits from grain are among the administrative measures that may partly have ameliorated negative drought impacts. Although it has been maintained

that hot, dry weather was often associated with locust outbreaks into central Europe, only in 1727–1728 were locusts reported in the Czech Lands during the selected years of extreme droughts (Brázdil et al., 2014).



Finally, extreme droughts elicited certain cultural responses. On the religious level, these were reflected in the organisation of processions of entreaty, featuring prayers for rain and thematic sermons, some of them published (AS5). "Creating drought" was also a charge levelled at people accused of witchcraft. A case in point was that of Tomáš Chvátal in 1681 (Rojčíková, 2000): "*If he wanted to stop*
*rain, he took spice, prepared a magic ointment and spread it in grease upon the breasts, three-times crossed himself, and then rain retreated.* […]. *Carrying three white pfennigs in the right shoe was another way of causing drought.*" In public space, extreme droughts could be recalled by folk/market songs. One example to survive is "A Key to the Rain, or a New Song for a Time of Drought" (*Klíč od deště aneb Nová píseň v čas sucha*), related to the drought of 1678, published in Prague in 1678 and a
year later (AS3). More substantial mementoes took the form of "hunger stones" in rivers (see Fig. 6).

The impacts and responses described may collated to constitute a model of ordered impacts, as proposed by Kates (1985) and documented more recently in greater detail by Dolák et al. (2015), Krämer (2015), and Luterbacher and Pfister (2015).

### 5.2.1 Extreme droughts and grain prices
A number of series of grain prices may be used as examples of how extreme Czech droughts were reflected in fluctuations of the cost of this basic commodity. Since it is impossible to find Czech grain prices series to cover the whole three centuries studied, available series from various parts of the period were used. Data concerning prices for the purchase and sale of wheat and rye in four selected Moravian
towns (Novotný, 1963) were used to create de-trended series expressing deviation from real grain prices for 1550–1621 (Fig. 7). The series differ in length: purchase of wheat in Brno (1570–1615) and Dačice (1576–1620) and rye in Znojmo (1580–1620), and sale of wheat in Olomouc (1550–1621) and Znojmo (1570–1621) and rye in Olomouc (1550–1615). The periods with available prices include a total of eight years with extreme drought episodes. Grain prices responded to drought by increasing in
1571 and 1590. However, in the latter year, the rises culminated a year later, after dry JJA and SON 1590. A slight increase in prices also appeared after dry SON 1580 and MAM 1603. Effects on prices were not evident in other extreme dry years, although significant rises, not related to droughts, occurred in 1600 and 1615.

Series of grain prices for Dačice (Brázdil and Duďáková, 2000) and Prague (Schebek, 1873)
may be used for the latter part of the 17th century and for the 18th century (Fig. 8). Fluctuations in grain prices show generally consistent variations for individual crops (wheat, rye, barley), as well as for the two places (Dačice, Prague). Many of the extremely dry MAM and JJA seasons are reflected in increases in grain prices for the same year or the following one. This holds for JJA 1630–1631, MAM 1638 (only wheat prices in Dačice are available these two), JJA 1684 and 1718–1719, MAM 1727 and
1732, JJA 1746 and MAM 1790. Dry JJA 1684 coincides with a steep increase in wheat prices, but only in the Dačice records. On the other hand, in the years 1683, 1753 and 1794, with extremely dry MAM, grain prices fell to local minima. An extremely dry MAM in 1800 is located in a sharply increasing gradient of grain prices of the late 1790s and the early 1800s, a period of general inflation that eventually resulted in monetary reform in 1811 (Štaif, 2017). A steep increase in grain prices in the
early 1770s is notable, also culminating in 1772. Harvest failure due to adverse weather in the years 1770–1771 combined with the socio-economic situation in the Czech Lands led to produce a great famine (known as "the hungry years"), with important economic, social and political consequences (Brázdil et al., 2001; Pfister and Brázdil, 2006).

Since the grain price series for Dačice and Prague were continuous, they were correlated with
reconstructed MAM, JJA and summer half-year series of SPI, SPEI and Z-index (Brázdil et al., 2016a). With the exceptions of MAM SPI and wheat prices in Dačice (correlation coefficient r = –0.16 significant at the 0.05 significance level), all the others were statistically insignificant (r = –0.17, the highest, was for barley prices). Despite this, a number of remarkable facts follow from comparisons of





correlations between the two places and between grain species (not shown). All three drought indices correlated best with wheat prices in Dačice and with barley prices in Prague. Wheat and rye prices generally correlated better with drought indices in Dačice, while barley prices exhibited slightly higher correlations in Prague.

The above results for the Czech Lands tally with a paper by Esper et al. (2017), in which grain prices series for 19 cities in central and southern Europe from the 14th to the 18th centuries were used to demonstrate possible environmental drivers of their fluctuations. They demonstrated that food shortages coincided with regional summer drought anomalies. Despite very low correlations of prices with tree-ring-based drought indices (r hardly exceeding –0.20), they showed that grain prices were

exceptionally high during dry periods.

Meteorological variables other than droughts influence harvests and yields of agricultural crops (grain in particular), which can again be reflected in grain prices. For example, Bauernfeind and Woitek (1999) analysed fluctuations in annual grain prices in the German cities of Nuremberg, Cologne, Augsburg and Munich for 1500–1599. They drew attention to the fact that the duration of the

vegetation period was an important factor in determining grain price fluctuations, reporting in particular a positive impact of precipitation in DJF and low SON temperatures on grain prices. Important increases in the impacts of climate on grain prices has been demonstrated for the climate deterioration in the second half of the 16th century (see Pfister and Brázdil, 1999). Brázdil and Durďáková (2000) analysed grain price series in the Moravian towns of Brno, Dačice and Olomouc in

the 16th–18th centuries and selected 61 years with extremely high prices for which the effects of weather and other factors on prices were studied. For two-thirds of these years, a relation of high prices to bad grain harvests in the given, or the preceding, year, was established, with adverse weather patterns.

However, general weather/climate effects on grain prices cannot be addressed separately, since

other disastrous weather events (e.g. hail, flood, frost) must be taken into account, as well as cereal pests and diseases. In particular, many socio-economic factors must be considered, not least among them wars, administrative decrees, corn reserves, expected grain yields, movement of grain in and out of the country, frequency of grain markets, and speculation (Petráň 1977). Prices have also been affected by the spatial extent of a given drought, with large-scale drought affecting several main

production regions simultaneously likely to have far greater effects than local drought. This may go some way towards explaining why some drought years (e.g. 1683, 1686 or 1779 and 1781) are not reflected in price increases. The relationship between drought and cereal yield may have changed over time, as shown by Trnka et al. (2012), who demonstrated that drier years in the late 19th century were associated in general with higher yields, probably due to the lower infestation pressure from pests and

diseases and more favourable harvest conditions. Similar trends have been reported on a European scale (Trnka et al., 2016a).

### 5.3 Extreme droughts in the pre-instrumental and instrumental periods

Future droughts in the context of recent climate change are a highly topical and controversial subject in

the light of their impacts on human society and its economic activities (e.g. Blauhut et al., 2016). Even the mental health of those affected may suffer (Vins et al., 2015). From this point of view, a comparison of extreme droughts from the pre-instrumental period (1501–1803) with those from the instrumental period (1804–2017) gains importance. Applying the same approach as that used for selection of extreme droughts for the pre-instrumental period in Sect. 4.1.2, extreme droughts in the

instrumental period were disclosed for only a few cases: one for MAM (1946 with 20-year return period of all three drought indices), six for JJA, seven for SON and five for the summer half-year (Table 3). Droughts with a return period of N ≥100 years detected by at least by one of three drought indices were identified in SON 1834, JJA and AMJJAS 1842, JJA 1904, AMJJAS 1947, SON 1953



and 1959, JJA 2003 and SON 2006. Only 1842, 1904 and 1911 for JJA, 1834, 1842, 1868 and 1947 for AMJJAS appeared among the ten driest years.

A lower number of extreme droughts in the 19th–21st centuries compared to the 16th–18th centuries may be related to a number of causes. Some may be an expression of natural climate variability. In general, periods of lower variability in drought index series are typified by a lower number of extremes (both positive and negative) and *vice versa* (Trnka et al., 2016b). Such periods may also occur in drought-sensitive proxies. For example, an oak tree-ring width chronology used for May–July precipitation reconstruction in Bohemia demonstrates a well-expressed period of more tree-ring width variability in the 16th and 17th centuries compared with recent times (see Fig. 2 in Dobrovolný et al., 2018). This may partly explain the higher number of extreme droughts in the "reconstruction" part of drought index series.

A higher number of extremes in the pre-instrumental period may also arise out of methodological concerns. Temperature and precipitation reconstructions (Dobrovolný et al., 2010, 2015) used for drought index calculations (Brázdil et al., 2016a) were based on simple linear regression with subsequent variance adjustment. Whereas the regression approach underestimates the variability of past climate (Esper et al., 2005), the variance adjustment partly overcomes this problem (McCarroll et al., 2004). However, variance adjustment is usually done for data from a relatively short overlapping period between target and proxy (1760–1854 in the case of temperatures and 1805–1854 in the case of precipitation in the above). This means that centuries-long reconstructions may underestimate or overestimate the real variability even after variance adjustment and thus produce more or fewer extremes compared with the entire instrumental period.

**6 Conclusions**

This contribution analysed selected extreme droughts in the Czech Lands during the pre-instrumental period (before AD 1804). The main conclusions can be summarised as follows:

(i) Calculation of return periods (N ≥20 years) in the reconstructed series of drought indices in the Czech Lands from the 1501–2017 period was used for selection of extreme droughts. This approach allows comparison of extreme droughts in both the pre-instrumental and instrumental periods. Selection of extreme droughts depends on the type of drought index and basic season.

(ii) Extreme droughts in the pre-instrumental period are characterised by significantly lower negative values of drought indices, higher temperatures and lower precipitation totals compared with the remainder of the years analysed. Seasonal composite of sea-level pressure (SLP) for extreme droughts gives significantly higher SLP values in the European region compared with the mean SLP in the 1961–1990 period.

(iii) Extreme droughts had a broad variety of impacts on human society and its responses. Among them, the most demonstrative influence on changes in grain prices depended not only on drought and its features, but also on other weather and socio-economic factors.

(iv) The number of extreme droughts derived from reconstructed series of drought indices was slightly higher in the pre-instrumental (16th –18th centuries) than in the instrumental period (19th–21st centuries). This may be partly related to natural climate variability or to the methods used for reconstruction of drought indices.

**Data availability.** The series of drought indices used in the paper for selection of extreme droughts and series of grain prices are available from the corresponding authors/publications. Sea-level pressure data may be obtained at https://www.ncdc.noaa.gov/paleo-search/study/6366.

**Competing interests.** The authors declare that they have no conflict of interest.





**Acknowledgements.** The authors acknowledge the financial support of the Czech Science Foundation for project no. 17-10026S. M.T. was supported by the SustES − Adaptation strategies for sustainable ecosystem services and food security under adverse environmental conditions project no. CZ.02.1.01/0.0/0.0/16_019/0000797. Tony Long (Svinošice) helped work up the English.

**Archival sources**

[AS1] Archiv města Ústí nad Labem, fond Sbírka rukopisů: Letopisecké záznamy Jana Čeledínka z Čáslavi připsané k Veleslavínovu Kalendáři historickému z r. 1590.

[AS2] Archiv Národního muzea v Praze, fond Sbírka rukopisů, sign. 288: Toto slovou registra bielá ab
anno 1539, sepsaná od Jana Jeníška z Újezda a na Svrčovci.

[AS3] Klíč od deště aneb Nová píseň v čas sucha. Prosba a klíč od deště, též za odvrácení hladu, moru, vojny i jiné potřeby celého křesťanstva ke cti Pánu Bohu Hospodáři Nebeskému, jejž on v své moci má, též dešťové nejsvětější Panně Marii Matce Boží Vyšehradské a svatým patronům českým na den svatého Václava léta 1678, když po velikém suchu spadla z nebe ponejprv vděčná vláha, na
poděkování toho i posavád trvajícího božího dobrodiní od Václava Šťastnýho Františka Rambeka, měštěnína N[ového] M[ěsta] P[ražského] k zpívání přihotovený. J[iž po] druhé vytištěný v Starém Městě Pražském u Daniele Michálka léta 1679.

[AS4] Knihovna Královské kanonie premonstrátů Praha-Strahov, sign. AG XII34: Anonymní záznamy počasí. In: Annorum priorum 30 Incipientium ab Anno Christi 1595, et definentium in annum 1624,
Ephemerides Brandenburgicae coelestium motuum et temporum. Summa diligentia in luminaribus calculo duplici Tychonico et Prutenico, in reliquis Planetis Prutenico seu Copernicaeo elaboratae, a Davide Origano Glacense Germano, Mathematico in Academia Electorali Brandenburgica Professore Publ. et ordinario. Typis excripsit Ioannes Eichorn Anno 1609. Apud Davidem Reichardum Bibliopolam Stetinensem.

[AS5] Městská knihovna Praha, sign. 35 D 19: O hrozném a velikém suchu zdržení dešťů a odtud následujícím nedostatku vody, jakéhož sucha žádný z lidí nynějších, ode sta let i výšeji, starých nepamatuje. Kázání učiněné v kostele domaželitském (sic!). Nyní pak kvůli pobožných křesťanů k probuzení lidu Božího, k horlivému pokání a modlitbám svatým vůbec vydané. Odemne (sic!) kněze Daniele staršího Philomatesa, služebníka Slova Páně v Domaželicích (sic!), Notariusa Řádu kněžstva
Páně evangelitského v podkraji (sic!) Holomouckém. Vytištěno v Holomoucí (sic!) u Jiříka Handle. Léta Páně 1616.

[AS6] Moravská zemská knihovna Brno, sign. A21: Hieronymus Haura, Miscellanea iucundo-curiosa in quibus continentur variae descriptiones, versus, carmina, elogia, epitaphia, vaticinia, illuminationes, declarationes, pugnae, conflictus, notata de bellis et diversis temporibus, casus laeto-fatales,
contingentia in monasterio Sancti Thomae, processiones et devotiones ad Thaumaturgam, varii eventus in Moravia, Bohemia, et adjacentibus regionibus, Brunae et aliis civitatibus, ac aliae iucundae, et utiles annotationes et reflexiones ... Quae omnia diligenter annotavit, laboriose conscripsit Pater Hieronymus Haura, Boemus Moldavo-Teynensis, Ord. Erem. D. P. Augustini, Brunae in Exempto Monasterio S. Thomae Professus ...

[AS7] Moravský zemský archiv Brno, fond E 55 Premonstráti Hradisko: Diaria kanonie Klášterní Hradisko 1693–1783, sign. II–18 (rok 1727), sign. II–19 (rok 1728).

[AS8] Moravský zemský archiv Brno, fond G 10 Sbírka rukopisů Zemského archivu 1200–1999, inv. č. 680: Jan Voldřich Klusák z Kostelce a na Radovesnicích, Historické a různé záznamy 1597–1689.



[AS9] Moravský zemský archiv Brno, fond G 11 Sbírka rukopisů Františkova muzea Brno 1300–1899, inv. č. 90: Josef Lucián Ondřej Kramoliš, Paběrky z dějou městečka Rožnova.

[AS10] Moravský zemský archiv Brno, fond G 13 Sbírka Historického spolku Brno 1306–1923, inv. č. 199: Kronika města Fulneku (Notizen über die Enstehung und Schicksale der Stadt Fulnek bis zum 5 Jahre 1806).

[AS11] Moravský zemský archiv Brno, fond G 13 Sbírka Historického spolku Brno 1306–1923, č. rkp. 432: Kronika Rýmařovska. 1405–1777.

[AS12] Národní muzeum Praha, sign. IV C 23: Řehoř Smrčka, Poznamenání některých pamětí od léta Páně 1587.

[AS13] Státní oblastní archiv Litoměřice, fond Velkostatek Mimoň – Stráž pod Rálskem, inv. č. 72: Pamětní kniha učitele Františka Tomáše Spillara z Plzeňska z let 1771–1907 s přípiskem jeho pokračovatele k r. 1844.

[AS14] Státní okresní archiv Blansko, pobočka Boskovice, fond Archiv městečka Olešnice, inv. č. 109: Kronika Jana Čupíka z Olešnice.

[AS15] Státní okresní archiv Česká Lípa, fond Sbírka rukopisů, sign. 13/3: Witterungs-Geschichte. Auszug aus den Titl: Lesenwürdige Sammlungen der hinterlegten Jahrgängen. Von Anton Lehmann Lehrer in Neuland. Abgeschrieben durch Joseph Meißner.

[AS16] Státní okresní archiv Litoměřice, fond Archiv města Litoměřice, sign. IV B 1a: Kniha pamětní litoměřických městských písařů 1570–1607.

[AS17] Státní okresní archiv Louny, fond AM Louny – kroniky, sign. Ch1: Chronica civitatis Launensisin Boemia Auctore Paulo Mikssowicz servo consulari.

[AS18] Státní okresní archiv Ústí nad Labem, sign. B.V.1/33: Usta ad Albim delinaeta carmine rebusque suis memorabilius illustrate, erga divis relligionis, erga patriam charissimam gratitudinis et observantes, ergo authore Joanne Augustino Tichtenbaum, patricio cive et senatore ibidem. Pragae, 25 apud haeredes Caspari Kargesii 1611.

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





Table 1. Overview of seasonal (MAM, JJA, SON) and summer half-year (AMJJAS) extreme droughts in the Czech Lands, selected according to series of SPI (a), SPEI (b) and Z-index (c) with a return period of N ≥20 years (N ≥10 years for SPI) in the pre-instrumental 1501–1803 period.

| MAM | | | | JJA | | | | SON | | | | AMJJAS | | | |
|---|---|---|---|---|---|---|---|---|---|---|---|---|---|---|---|
| Year | a | b | c | Year | a | b | c | Year | a | b | c | Year | a | b | c |
| 1532 | 50 | 20 | 20 | 1503 | 10 | 20 | 20 | 1536 | 10 | 20 | 50 | 1504 | 10 | 20 | 20 |
| 1540 | 50 | 200 | 100 | 1504 | 10 | 20 | 20 | 1540 | 20 | 100 | 200 | 1534 | 10 | 20 | 20 |
| 1571 | 20 | 50 | 50 | 1534 | 10 | 20 | 20 | 1548 | 100 | 50 | 20 | 1536 | 20 | 50 | 50 |
| 1583 | 20 | 20 | 20 | 1536 | 20 | 50 | 50 | 1580 | 100 | 20 | 20 | 1540 | 200 | 200 | 200 |
| 1603 | 50 | 50 | 50 | 1540 | 200 | 200 | 200 | 1590 | 20 | 50 | 100 | 1590 | 200 | 100 | 100 |
| 1638 | 200 | 200 | 200 | 1556 | 20 | 20 | 20 | 1605 | 100 | 100 | 50 | 1616 | 100 | 100 | 200 |
| 1683 | 200 | 50 | 20 | 1590 | 100 | 200 | 200 | 1631 | 10 | 20 | 20 | 1631 | 20 | 20 | 50 |
| 1686 | 20 | 50 | 20 | 1616 | 50 | 100 | 200 | 1634 | 200 | 50 | 50 | 1684 | 10 | 20 | 20 |
| 1727 | 50 | 50 | 20 | 1630 | 50 | 20 | 20 | 1680 | 100 | 200 | 200 | 1706 | 10 | 20 | 20 |
| 1732 | 20 | 20 | 20 | 1631 | 10 | 20 | 20 | 1686 | 20 | 20 | 50 | 1718 | 10 | 20 | 20 |
| 1753 | 20 | 20 | 20 | 1666 | 20 | 20 | 20 | 1710 | 20 | 20 | 20 | 1726 | 10 | 20 | 20 |
| 1779 | 200 | 200 | 200 | 1684 | 20 | 50 | 50 | 1726 | 20 | 20 | 50 | 1727 | 20 | 59 | 50 |
| 1781 | 50 | 20 | 20 | 1718 | 20 | 20 | 20 | 1727 | 20 | 50 | 100 | 1728 | 10 | 20 | 100 |
| 1790 | 200 | 20 | 20 | 1719 | 20 | 20 | 20 | 1731 | 10 | 20 | 20 | 1800 | 20 | 20 | 20 |
| 1794 | 50 | 200 | 200 | 1728 | 20 | 20 | 100 | 1754 | 10 | 20 | 20 | | | | |
| 1800 | 50 | 50 | 50 | 1746 | 100 | 50 | 50 | 1772 | 20 | 20 | 20 | | | | |



Table 2. Comparison of selected extreme years for the Czech Lands from drought indices based on documentary data (DD) in this paper with those selected from AMJJA SPEI based on grape harvest dates (GHD) in Bohemia, MAMJJ precipitation in South Moravia based on fir tree-rings (TR-1), MJJ precipitation in Bohemia based on oak tree-rings (TR-2), JJA scPDSI in the Czech Republic (TR-CZ)

5 based on tree-rings and JJA scPDSI in central Europe (TR-CE) based on tree-rings.

| Data | Variable | Source | Extreme drought | |
|---|---|---|---|---|
| | | | Order 1–10 | Order 10–20 |
| DD | AMJJAS SPI, SPEI, Z-index | This paper | 1540, 1590, 1616, 1727, 1536, 1631 | 1800, 1728, 1534, 1726, 1718, 1504, 1706, 1684, 1794 |
| DD | MAM SPI, SPEI, Z-index | This paper | 1638, 1779, 1540, 1794, 1603, 1800, 1683, 1727, 1571, 1781 | 1790, 1732, 1532, 1723, 1728, 1616, 1686, 1541 |
| DD | JJA SPI, SPEI, Z-index | This paper | 1540, 1590, 1616, 1746, 1536, 1684, 1728 | 1666, 1718, 1719, 1556, 1630, 1534, 1504, 1615 |
| GHD | AMJJA SPEI | Možný et al. (2016) | 1540, 1616, 1590 | 1706, 1718, 1536, 1556, 1719 |
| TR-1 | MAMJJ precipitation | Brázdil et al. (2002) | 1653, 1636, 1561, 1603, 1790, 1525 | 1779, 1616 |
| TR-2 | MJJ precipitation | Dobrovolný et al. (2018) | 1616, 1503, 1532, 1779, 1525, 1564, 1541, 1502, 1538 | 1800, 1603, 1790, 1567 |
| TR-CZ | JJA scPDSI | Cook et al. (2015) | 1616, 1624, 1517, 1503, 1525 | 1504, 1540, 1636, 1603, 1718 |
| TR-CE | JJA scPDSI | Cook et al. (2015) | 1503, 1504, 1684, 1517 | 1636, 1616, 1784, 1653, 1540, 1719, 1624, 1718 |





Table 3. Overview of seasonal (JJA, SON) and summer half-year (AMJJAS) extreme droughts in the Czech Lands selected according to series of SPI (a), SPEI (b) and Z-index (c) with a return period of N ≥20 years (N ≥10 years for SPI) in the instrumental 1804–2017 period.

| JJA | | | | SON | | | | AMJJAS | | | |
|---|---|---|---|---|---|---|---|---|---|---|---|
| Year | a | b | c | Year | a | b | c | Year | a | b | c |
| 1842 | 200 | 50 | 50 | 1834 | 50 | 50 | 100 | 1834 | 50 | 20 | 20 |
| 1868 | 20 | 20 | 20 | 1865 | 10 | 20 | 20 | 1842 | 200 | 50 | 50 |
| 1904 | 200 | 50 | 20 | 1947 | 10 | 20 | 50 | 1868 | 50 | 50 | 20 |
| 1911 | 50 | 20 | 20 | 1953 | 200 | 100 | 50 | 1947 | 50 | 100 | 100 |
| 2003 | 20 | 100 | 100 | 1959 | 200 | 100 | 50 | 2003 | 10 | 50 | 20 |
| 2015 | 10 | 20 | 20 | 1982 | 20 | 50 | 20 | | | | |
| | | | | 2006 | 20 | 100 | 100 | | | | |




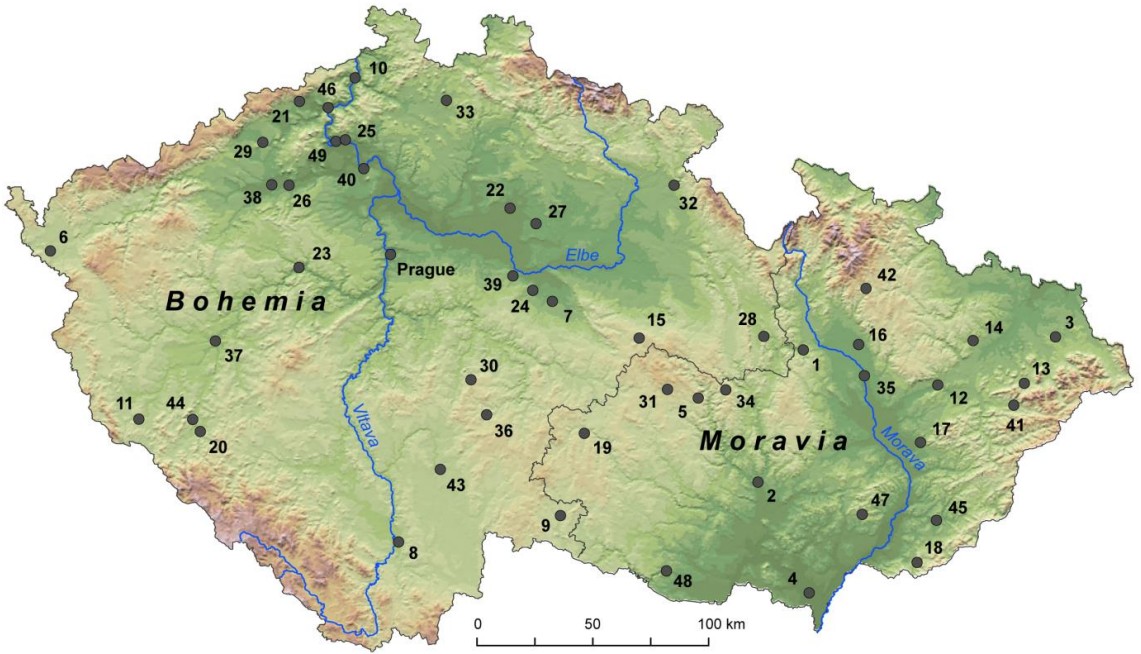

Figure 1. Location of places reported in the recent Czech Republic: 1 – Bouzov, 2 – Brno, 3 – Bruzovice, 4 – Břeclav, 5 – Bystřice nad Pernštejnem, 6 – Cheb, 7 – Čáslav, 8 – České Budějovice, 9 – Dačice, 10 – Děčín, 11 – Domažlice, 12 – Drahotuše, 13 – Frenštát pod Radhoštěm, 14 – Fulnek, 15 – Hlinsko, 16 – Hnojice, 17 – Holešov, 18 – Javorník, 19 – Jihlava, 20 – Klatovy, 21 – Krupka, 22 – Křinec, 23 – Křivoklát, 24 – Kutná Hora, 25 – Litoměřice, 26 – Louny, 27 – Městec Králové, 28 – Moravská Třebová, 29 – Most, 30 – Načeradec, 31 – Nové Město na Moravě, 32 – Nové Město nad Metují, 33 – Noviny pod Ralskem, 34 – Olešnice, 35 – Olomouc, 36 – Pacov, 37 – Plzeň, 38 – Postoloprty, 39 – Radovesnice, 40 – Roudnice nad Labem, 41 – Rožnov pod Radhoštěm, 42 – Rýmařov, 43 – Soběslav, 44 – Svrčovec, 45 – Uherský Brod, 46 – Ústí nad Labem, 47 – Vřesovice, 48 – Znojmo, 49 – Žalhostice.





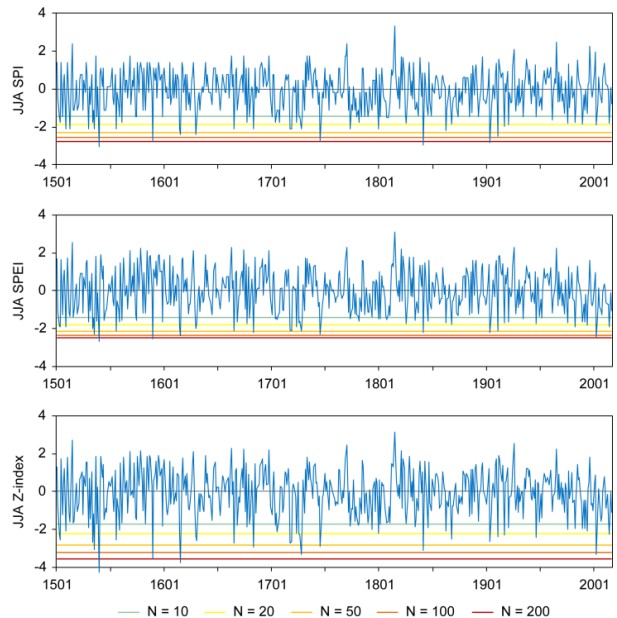

Figure 2. Fluctuations in JJA SPI, JJA SPEI and JJA Z-index in the Czech Lands in the 1501–2017
period. Coloured horizontal lines mark the limits for estimation of extreme droughts with return periods
of N = 10, 20, 50, 100, and 200 years.

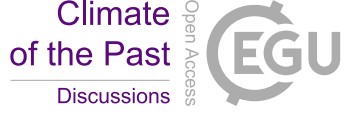



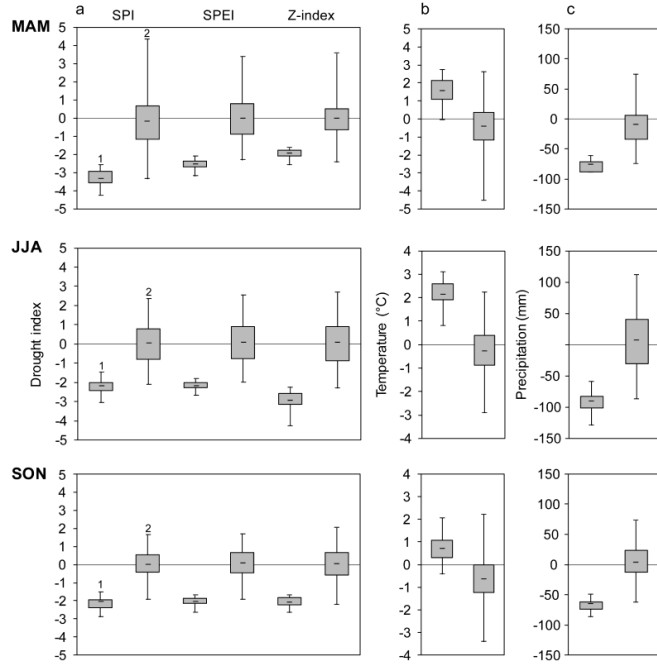

Figure 3. Box-plots (maximum and minimum, upper and lower quartile, mean) of Czech drought indices (a), central European temperatures (b) and Czech precipitation (c) in MAM, JJA and SON: 1 – dry years with N ≥20 years, 2 – remaining years 1501–1803. Temperatures and precipitation are expressed as anomalies with respect to the 1961–1990 reference period.



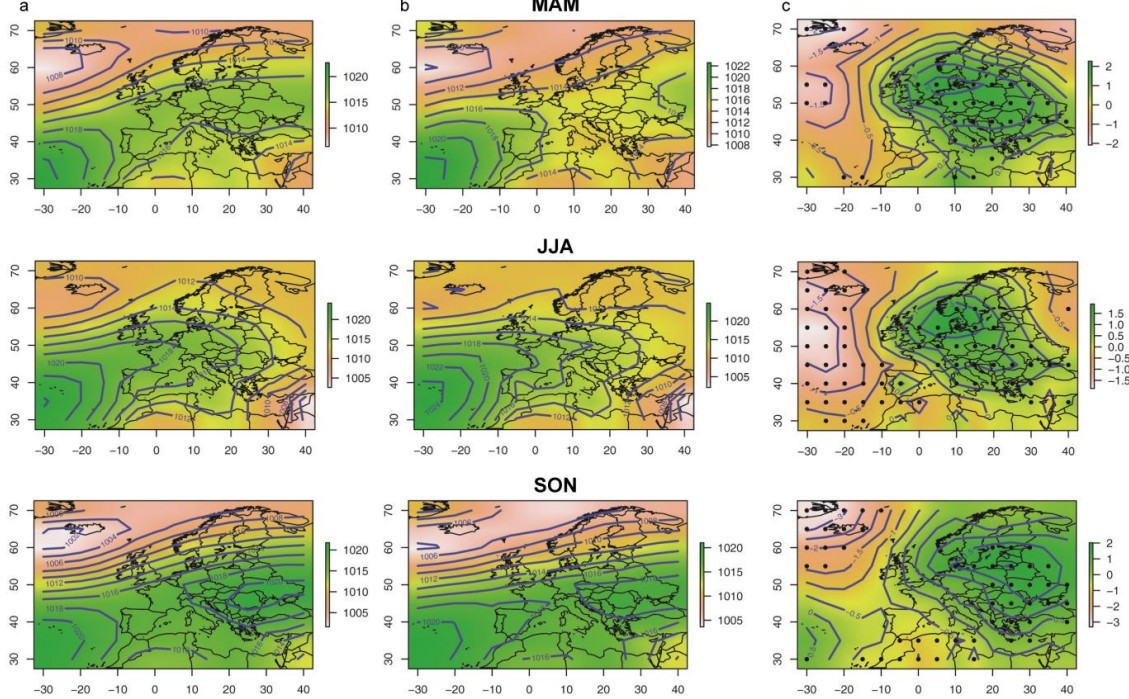

Figure 4. MAM, JJA and SON SLP of Czech extreme droughts in the Atlantic-European region: a) SLP composite of extreme Czech droughts, b) mean SLP field in the 1961–1990 period, c) seasonal differences of SLP composite and mean SLP (dots identify grids in which pressure differences were statistically significant).

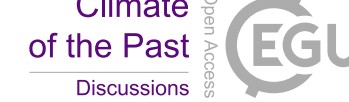



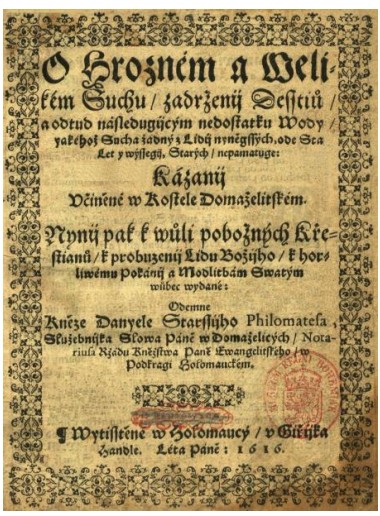

Figure 5. Printed sermon by the Reverend Daniel Philomates the Elder (AS5) related to the severe
drought of 1616 in the Czech Lands (from the collection of the National Museum, Prague).



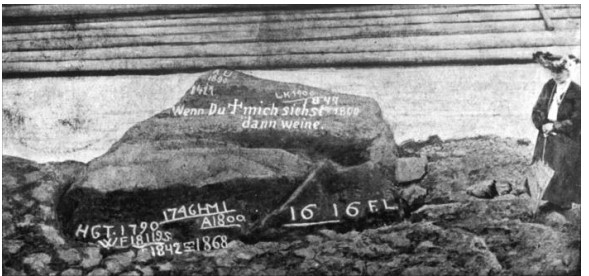

Figure 6. The hunger stone that appeared during the severe 1904 drought on the left bank of the River
Elbe at Děčín-Podmokly, recording certain low water levels and hydrological droughts in Bohemia
(O. Kotyza archive).



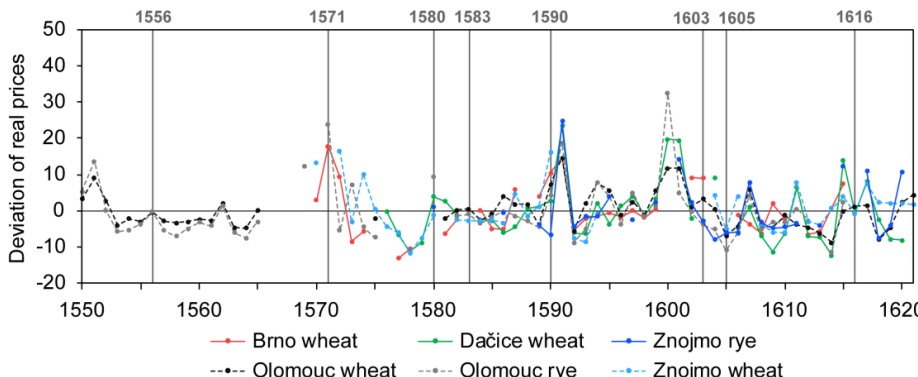

Figure 7. Deviations of real prices of purchase (full line) and sale (broken line) of wheat and rye in Brno, Dačice, Olomouc and Znojmo in the 1550–1621 period, with identification of years of extreme
5  drought.





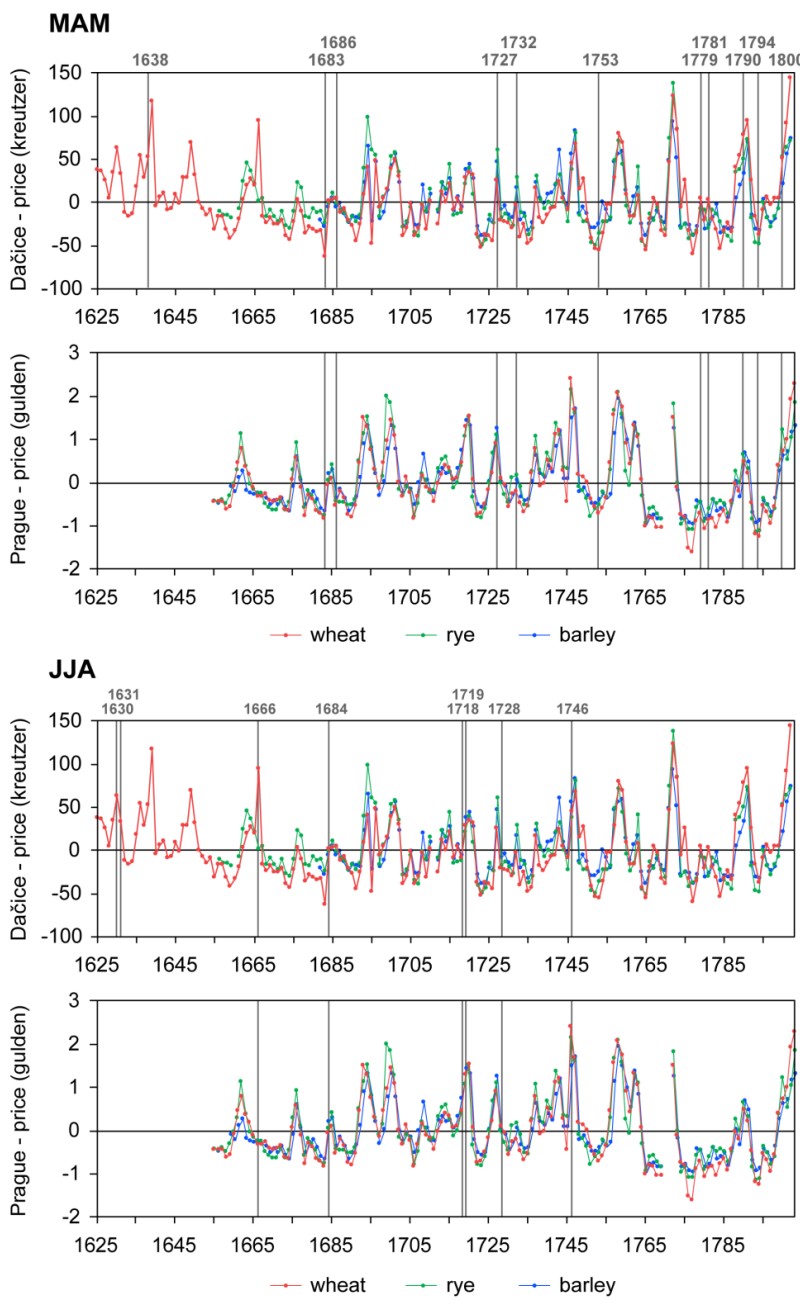

Figure 8. Deviations in grain prices (wheat, rye and barley) in Dačice for 1625–1803 (Brázdil and Durďáková, 2000) and in Prague in 1655–1803 (Schebek, 1873) with identification of extremely dry MAM and JJA seasons.