# Peer review of "Extreme droughts and human responses to them: the Czech Lands in the pre-instrumental period"

_Climate of the Past, 2018_

## Referee Comment (RC1) · D. Nash (Referee) · 28 Oct 2018

**General comments**

Thank you for the opportunity to review this manuscript, which presents a detailed analysis of extreme droughts in the Czech Lands during the 12th-18th centuries. It is extremely well-written, founded on a rich empirical basis, and demonstrates clear and systematic analysis. It makes some interesting conclusions that will be of considerable relevance for other analysis of historical droughts. I was particularly struck by the observations on p.12 regarding the identification (or otherwise) of spatially limited droughts, which could only be made from such a rich dataset. I have no hesitation in recommending it for publication, subject to a few recommendations.

[Figure]

Specific comments

1. The introduction to the paper is generally good, and sets a sound context for the study. However, it could stress the novelty of the investigation more forcefully. There are relatively few parts of the world where there is a sufficiently rich documentary dataset to conduct this type of study. This should come through more forcefully in p.2 lines 29-40.

2. I found the text in sub-section 4.1.1 of Sect. 4.1 rather confusing. The overall heading for section 4.1 is 'Extreme droughts in the Czech Lands during the pre-instrumental period' but the text in section 4.1.1 (p.5 line 29-30) indicates that 'it is difficult to identify cases that could clearly be classified as extreme'. So are these droughts before AD 1500 extreme? If not, then why is the paragraph a subset of section 4.1? I suggest the authors either drop the subheadings to 4.1.1 and 4.1.2 and group the text together, or move lines 21-30 to immediately after the heading to section 4 (i.e. before 4.1) as an introduction to the whole section.

3. Sections 4.2.1 and 4.2.2, while rich with detail, take up 4.5 pages of the 15.5 pages of text in the manuscript. I wouldn't suggest losing these sections, but (depending on the policy of Climate of the Past) they could be more usefully moved to the end of the document as an appendix.

4. It would help the reader (and help link the descriptions of droughts in sections 4.2.1 and 4.2.2 to the rest of the manuscript) if the text in section 5.2 could cross-reference to specific drought years as examples. So, the mention of famine in p.13 line 42 could note years when famine was identified in parentheses. I'm not suggesting that all droughts need to be cross-referenced, but mentions of noteworthy examples would bring the sections of the manuscript together.

5. Tables 1 and 3 could be made more clear for the reader. I would recommend replacing 'a, b, c' in the column headings of Tables 1 and 3 with 'SPI, SPEI and Z-index', such that the reader does not need to cross-refer with the table caption.

6. I found Table 2 very difficult to interpret – could this be made more user-friendly? I assume that I am meant to read it by looking down the columns 'Order 1-10' and 'Order 1-20' to see which years appear in multiple studies? This isn't easy across multiple centuries. Could the authors perhaps highlight years that appear in the majority of series (e.g. by putting them in bold or italics) or think of a graphical way to display the data?

7. I found it difficult to compare the three parts of Figure 2. Could the diagram be made larger and more easily readable (e.g. by inserting some vertical lines at each 0/100 year tick mark)?

Technical corrections

p.1 line 37 – '. . .in a deficit...' (delete 'a')

p.1 line 39 – '. . .this may be exacerbated. . .' (insert 'be')

p.2 line 27 – '. . .in the other areas of the world. . .' (delete first 'the')

p.2 line 29 – '. . .already exist for the. . .' (insert 'for')

p.2 line 32 – 'However, the somewhat. . .' (delete 'However', capitalise 'The')

p.2 line 44 – '. . .may be used for the. . .' (insert 'the')

p.2 line 45-46 – replace '. . .mean monthly precipitation series calculated for the territory of the Czech Lands' with '. . .mean monthly calculated precipitation series'.

p.3 line 14 – 'On the other hand, records related. . .' (delete 'On the other hand', capitalise 'Records')

p.3 line 22 – not sure what 'nad Labem' means.

p.4 line 37 – ' and the instrumental periods. . .' (delete 'the')

p.4 line 39 – 'On the other hand, the. . .' (delete 'On the other hand', capitalise 'The')

p.4 line 39 – replace '. . .may be taken as a certain disadvantage' with '. . .may be considered as a minor disadvantage'

p.5 line 35 – '. . . that the use of. . .' (delete 'the')

p.13 line 1 – replace 'Impacts on social life and human responses. . .' with 'Impacts on society and human responses. . .'

p.13 line 13 – note sure what 'aftermath harvest' means – do you mean 'poor ensuing harvest'?

p.13 line 13 – '. . . had to be bought. . .' (insert 'be')

p.13 line 21 – replace 'eruptions' with 'outbreaks'?

p.13 line 22 – '. . .rendering them small, maggoty. . .' (delete comma after 'them')

p.13 line 25 – replace 'if fire that broke out. . .' with 'if fire also broke out. . .'

p.14 line 11 – '. . . may be collated. . .' (insert 'be')

p.14 line 15 – should section 5.2.1 not be section 5.3? There isn't a section 5.2.2. If so, section 5.3 will need renumbering to 5.4

p.14 line 41 – '. . .led to produce a great. . .' (delete 'produce')
* * *

---

## Referee Comment (RC2) · C. Rohr (Referee) · 13 Nov 2018

Chapter 4 contains a very useful overview of the strongest droughts based on documentary evidence. However, it does not always come clear, whether these reports are fully reliable, because they are contemporary (or even written by eyewitnesses) or not. Some of the sources in the bibliography, but some are not (e.g. AS6: Hieronymus Haura). It will be useful for historians in particular to add a short information concerning contemporary or not in the text.

Chapter 4.2.2.3: I am not really sure, if we can deduce an autumn drought from this relatively poor documentary evidence. As far as we know, summer 1548 was very dry. In this way, low water in late autumn may also result from this period combined with an

at least relatively dry autumn, so that people could cross the riverbed of the Elbe River in early December.

Chapter 5.2 is a very important part of the discussion chapter. Maybe you could also add one or two sentences (p. 13, l. 20 sqq.) on the ambivalent consequences of droughts towards fruit production. Whereas fruit trees (apples etc.) were obviously affected by caterpillars or the like, wine was growing even better sometimes, as you show for 1503, 1536, 1540 etc. However, I would also appreciate if you could add some information if there were any learning processes to prevent similar shortages after droughts, e.g. by installing or enlarging granaries (or mention that the sources do not tell us much about prevention).

Please see some minor corrections of typos in the bibliography mostly concerning titles in Latin and German.

Please also note the supplement to this comment:
https://www.clim-past-discuss.net/cp-2018-135/cp-2018-135-RC2-supplement.pdf

---

## Referee Comment (RC3) · C. Rohr (Referee) · 14 Nov 2018

Christian Rohr: Comment on Brázdil, R. et al: Extreme droughts and human responses to them: the Czech Lands in the pre-instrumental period (Climate of the past – Discussions)

This is a very important and rich contribution to historical droughts in Central Europe based on the long tradition of research on weather and climate in the Czech Lands by Rudolf Brázdil and his group. The article is on a very high methodological level by combining a large number of documentary evidence with the most important drought indices (SPI, SPEI, Z-index). The authors testify an excellent overview of the state of the art (see also the large bibliography). In the discussion chapter, they differentiate

very well when combining droughts with grain prices (which could be influenced by various factors).

Chapter 4 contains a very useful overview of the strongest droughts based on documentary evidence. However, it does not always come clear, whether these reports are fully reliable, because they are contemporary (or even written by eyewitnesses) or not. Some of the sources in the bibliography, but some are not (e.g. AS6: Hieronymus Haura). It will be useful for historians in particular to add a short information concerning contemporary or not in the text.

Chapter 4.2.2.3: I am not really sure, if we can deduce an autumn drought from this relatively poor documentary evidence. As far as we know, summer 1548 was very dry. In this way, low water in late autumn may also result from this period combined with an at least relatively dry autumn, so that people could cross the riverbed of the Elbe River in early December.

Chapter 5.2 is a very important part of the discussion chapter. Maybe you could also add one or two sentences (p. 13, l. 20 sqq.) on the ambivalent consequences of droughts towards fruit production. Whereas fruit trees (apples etc.) were obviously affected by caterpillars or the like, wine was growing even better sometimes, as you show for 1503, 1536, 1540 etc. However, I would also appreciate if you could add some information if there were any learning processes to prevent similar shortages after droughts, e.g. by installing or enlarging granaries (or mention that the sources do not tell us much about prevention).

Please see some minor corrections of typos in the bibliography mostly concerning titles in Latin and German. In addition, as also mentioned by the second reviewer (an English native speaker), the text needs some more corrections on language and style by a native speaker (e.g. missing articles, construction of some sentences).

In general, this is an important contribution, which should be accepted with minor revision, i.e. there are mostly some technical improvements (language, typos) and some clarifications needed, as mentioned in my comments.

Please also note the supplement to this comment:
https://www.clim-past-discuss.net/cp-2018-135/cp-2018-135-RC3-supplement.pdf

―――――――――――――――――

**Supplement:**

[revised manuscript text omitted]

---

## Author Comment (AC1) · 8 Dec 2018

D. Nash (Referee) d.j.nash@brighton.ac.uk

General comments Thank you for the opportunity to review this manuscript, which presents a detailed analysis of extreme droughts in the Czech Lands during the 12th-18th centuries. It is extremely well-written, founded on a rich empirical basis, and demonstrates clear and systematic analysis. It makes some interesting conclusions that will be of considerable relevance for other analysis of historical droughts. I was particularly struck by the observations on p.12 regarding the identification (or other-

wise) of spatially limited droughts, which could only be made from such a rich dataset. I have no hesitation in recommending it for publication, subject to a few recommendations. RESPONSE: We would like to thank D. Nash for the generally positive evaluation of our manuscript and the number of constructive comments/suggestions, which we address below.

Specific comments 1. The introduction to the paper is generally good, and sets a sound context for the study. However, it could stress the novelty of the investigation more forcefully. There are relatively few parts of the world where there is a sufficiently rich documentary dataset to conduct this type of study. This should come through more forcefully in p.2 lines 29-40. RESPONSE: Accepted, following sentences were added into the quoted paragraph as follows: "Although a number of studies of droughts based on documentary evidence already exist for the Czech Lands (the recent Czech Republic) (Munzar, 2004; Brázdil et al., 2013; Brázdil and Trnka, 2015; Munzar and Ondráček, 2016), the current investigation concentrates on the comprehensive study of extreme droughts in the pre-instrumental period from the 12th to the 18th centuries. The somewhat episodic character of drought information before AD 1500 dictates that the primary focus is confined to extreme droughts during the 16th to the 18th centuries. This type of study is made possible by the wealth of historical documentary evidence, reaching back several centuries that exists in the Czech Lands. This body of evidence has now been researched, collected and collated for nearly three decades. A particularly novel feature of this study is also that it constitutes an "objective" selection of extreme droughts based on long-term series of drought indices reconstructed from such documentary data."

2. I found the text in sub-section 4.1.1 of Sect. 4.1 rather confusing. The overall heading for section 4.1 is 'Extreme droughts in the Czech Lands during the pre-instrumental period' but the text in section 4.1.1 (p.5 line 29-30) indicates that 'it is difficult to identify cases that could clearly be classified as extreme'. So are these droughts before AD 1500 extreme? If not, then why is the paragraph a subset of section 4.1? I suggest the

authors either drop the subheadings to 4.1.1 and 4.1.2 and group the text together, or move lines 21-30 to immediately after the heading to section 4 (i.e. before 4.1) as an introduction to the whole section. RESPONSE: Accepted and corrected. We deleted the subheadings of sections 4.1.1 and 4.1.2 and grouped it together.

3. Sections 4.2.1 and 4.2.2, while rich with detail, take up 4.5 pages of the 15.5 pages of text in the manuscript. I wouldn't suggest losing these sections, but (depending on the policy of Climate of the Past) they could be more usefully moved to the end of the document as an appendix. RESPONSE: Sorry, but we totally disagree with this proposal. Presentation of outstanding droughts in Section 4.2 we see as an inseparable part of our results. Finding and interpretation of documentary data represents very important part of research work in historical climatology. Moreover, we see publication of these results as important for further researchers trying to compare their droughts found in a broader territorial scale. If we remove Sections 4.2.1 and 4.2.2 into an appendix what will remain from our Sect. 4 Results? It would be a total degradation of the whole article. We do not know any policy of Climate of the Past supporting this proposal of the referee.

4. It would help the reader (and help link the descriptions of droughts in sections 4.2.1 and 4.2.2 to the rest of the manuscript) if the text in section 5.2 could cross-reference to specific drought years as examples. So, the mention of famine in p.13 line 42 could note years when famine was identified in parentheses. I'm not suggesting that all droughts need to be cross-referenced, but mentions of noteworthy examples would bring the sections of the manuscript together. RESPONSE: Accepted and corrected. We changed a last sentence of the first paragraph ("Reported below are only those drought impacts and responses that are known from Czech documentary evidence (with cross-referenced examples of specific extreme dry years mentioned in Sect. 4.2).") and complemented corresponding years of extreme droughts in many places of the manuscript in Sect. 5.2.

5. Tables 1 and 3 could be made more clear for the reader. I would recommend

replacing 'a, b, c' in the column headings of Tables 1 and 3 with 'SPI, SPEI and Zindex', such that the reader does not need to cross-refer with the table caption. RESPONSE: Accepted and corrected. It was a practical reason to use 'a, b, c' instead of 'SPI, SPEI and Z-index' to accommodate all data into both tables in any reasonable form. Now we changed their headings as requested.

6. I found Table 2 very difficult to interpret – could this be made more user-friendly? I assume that I am meant to read it by looking down the columns 'Order 1-10' and 'Order 1-20' to see which years appear in multiple studies? This isn't easy across multiple centuries. Could the authors perhaps highlight years that appear in the majority of series (e.g. by putting them in bold or italics) or think of a graphical way to display the data? RESPONSE: We complemented Table 2 including all extreme droughts of order 1–10 and 11–20 for all series used, when cases in the pre-instrumental period are expressed in bold. It allows now to do a detail comparison our selection of extreme droughts with those from other reconstructions. What we see as important message from this table for a reader we included in the corresponding paragraph in Sect. 5.2 as follows: "As might be expected, all the extreme droughts reconstructed for the Bohemian wine-growing region (Možná et al., 2016) are also reflected in selected droughts in the current paper (Table 2). From 13 pre-instrumental droughts based on oak tree-rings in Bohemia (Dobrovolná et al., 2018), agreement with those in the current paper occurs in eight cases. Three remaining tree-ring-based series exhibit agreement with them in a half the cases. If the 20 driest years of all eight series are taken and analysed together, only the year 1616 appears in all of them. The topical discussion of a "megadrought" in 1540 by Büntgen et al. (2015) and Pfister et al. (2015) appears justified, since this event was the most severe in all the documentary-based datasets, except in MAM, where 1540 was the third most severe. While the megadrought does not appear among the 20 most severe cases only in two Czech tree-ring series (Brázdil et al., 2002; Dobrovolná et al., 2018), it corresponds to the 12th order for the territory of the Czech Republic and to the 17th order for central Europe in series derived from OWDA (Cook et al., 2015). The year 1718 appeared among the driest months in five

series and the years of 1504 and 1603 in four series. Many extreme dry years are detectable either in documentary-based series or tree-ring based series." We believe that there is not necessary to include further details which can be further derived from Table 2.

7. I found it difficult to compare the three parts of Figure 2. Could the diagram be made larger and more easily readable (e.g. by inserting some vertical lines at each 0/100 year tick mark)? RESPONSE: Accepted, we added vertical lines for each 100-year interval. We would like only show fluctuation of a given drought index with respect to calculated thresholds, i.e. the figure has only an illustrative character. We do not expect that the reader of the article will compare its three parts (corresponding selected extreme droughts are in Table 1). The final size of the figure will be depending on a technical elaboration of the article in case it will be accepted for publication.

Technical corrections p.1 line 37 – ': : :in a deficit...' (delete 'a') p.1 line 39 – ': : :this may be exacerbated: : :' (insert 'be') p.2 line 27 – ': : :in the other areas of the world: : :' (delete first 'the') p.2 line 29 – ': : :already exist for the: : :' (insert 'for') p.2 line 32 – 'However, the somewhat: : :' (delete 'However', capitalise 'The') p.2 line 44 – ': : :may be used for the: : :' (insert 'the') p.2 line 45-46 – replace ': : :mean monthly precipitation series calculated for the territory of the Czech Lands' with ': : :mean monthly calculated precipitation series'. p.3 line 14 – 'On the other hand, records related: : :' (delete 'On the other hand', capitalise 'Records') RESPONSE: Accepted and corrected.

p.3 line 22 – not sure what 'nad Labem' means. RESPONSE: A full Czech geographic name is Roudnice nad Labem.

p.4 line 37 – ' and the instrumental periods: : :' (delete 'the') p.4 line 39 – 'On the other hand, the: : :' (delete 'On the other hand', capitalise 'The') p.4 line 39 – replace ': : :may be taken as a certain disadvantage' with ': : :may be considered as a minor disadvantage' p.5 line 35 – ': : : that the use of: : :' (delete 'the') p.13 line 1 – replace 'Impacts on social life and human responses: : :' with 'Impacts on society and human

responses: : :' RESPONSE: Accepted and corrected.

p.13 line 13 – note sure what 'aftermath harvest' means – do you mean 'poor ensuing harvest'? RESPONSE: This is a term which is used for the second haymaking (the first haymaking is usually in June, while in this case "aftermath" is harvested usually at the end of August or in the early September).

p.13 line 13 – ': : : had to be bought: : :' (insert 'be') p.13 line 21 – replace 'eruptions' with 'outbreaks'? p.13 line 22 – ': : :rendering them small, maggoty: : :' (delete comma after 'them') p.13 line 25 – replace 'if fire that broke out: : :' with 'if fire also broke out: : :' p.14 line 11 – ': : : may be collated: : :' (insert 'be') RESPONSE: Accepted and corrected.

p.14 line 15 – should section 5.2.1 not be section 5.3? There isn't a section 5.2.2. If so, section 5.3 will need renumbering to 5.4 RESPONSE: No, we would like to preserve this numbering of sections. The reason is that Section 5.2 has more general title (and content) "Impacts on society and human responses to extreme droughts", while "5.2.1 Extreme droughts and grain prices" is devoted only to one particular possible impact, i.e. only to effects of extreme droughts on grain prices.

p.14 line 41 – ': : :led to produce a great: : :' (delete 'produce') RESPONSE: Accepted and corrected.

---

## Author Comment (AC2) · 8 Dec 2018

C. Rohr (Referee) christian.rohr@hist.unibe.ch

Chapter 4 contains a very useful overview of the strongest droughts based on documentary evidence. However, it does not always come clear, whether these reports are fully reliable, because they are contemporary (or even written by eyewitnesses) or not. Some of the sources in the bibliography, but some are not (e.g. AS6: Hieronymus Haura). It will be useful for historians in particular to add a short information concerning

contemporary or not in the text.

Chapter 4.2.2.3: I am not really sure, if we can deduce an autumn drought from this relatively poor documentary evidence. As far as we know, summer 1548 was very dry. In this way, low water in late autumn may also result from this period combined with an at least relatively dry autumn, so that people could cross the riverbed of the Elbe River in early December.

Chapter 5.2 is a very important part of the discussion chapter. Maybe you could also add one or two sentences (p. 13, l. 20 sqq.) on the ambivalent consequences of droughts towards fruit production. Whereas fruit trees (apples etc.) were obviously affected by caterpillars or the like, wine was growing even better sometimes, as you show for 1503, 1536, 1540 etc. However, I would also appreciate if you could add some information if there were any learning processes to prevent similar shortages after droughts, e.g. by installing or enlarging granaries (or mention that the sources do not tell us much about prevention). Please see some minor corrections of typos in the bibliography mostly concerning titles in Latin and German. Please also note the supplement to this comment: https://www.clim-past-discuss.net/cp-2018-135/cp-2018-135-RC2-supplement.pdf

RESPONSE: Because these comments represent incomplete version of the review by C. Rohr submitted a day later, we replay to his comments completely in our responses to RC3.

---

## Author Comment (AC3) · 8 Dec 2018

C. Rohr (Referee) christian.rohr@hist.unibe.ch

Christian Rohr: Comment on Brázdil, R. et al: Extreme droughts and human responses to them: the Czech Lands in the pre-instrumental period (Climate of the past – Discussions) This is a very important and rich contribution to historical droughts in Central Europe based on the long tradition of research on weather and climate in the Czech Lands by Rudolf Brázdil and his group. The article is on a very high methodological

level by combining a large number of documentary evidence with the most important drought indices (SPI, SPEI, Z-index). The authors testify an excellent overview of the state of the art (see also the large bibliography). In the discussion chapter, they differentiate very well when combining droughts with grain prices (which could be influenced by various factors). RESPONSE: We would like to thank C. Rohr for a carefull review and generally positive evaluation of our manuscript and the number of constructive comments/suggestions, which we address below (our responses and changes compared to the original manuscript are in red colour).

Chapter 4 contains a very useful overview of the strongest droughts based on documentary evidence. However, it does not always come clear, whether these reports are fully reliable, because they are contemporary (or even written by eyewitnesses) or not. Some of the sources in the bibliography, but some are not (e.g. AS6: Hieronymus Haura). It will be useful for historians in particular to add a short information concerning contemporary or not in the text. RESPONSE: To avoid doubts about use of primary sources, we changed the first sentence in Sect. 2.1 Documentary data of droughts as follows: "A variety of primary documentary sources may be used for the identification of droughts in the pre-instrumental period in the Czech Lands, i.e. before AD 1804, ..." In the whole paper we are working particularly with primary sources, but their quotation is different (see lines 29–31, page 6 in the original manuscript). If such data were already critically elaborated and published, then we quote corresponding publication (see References), what seems to us to be more convenient for readers to find corresponding reports than to search in original manuscripts located in archives, libraries, in private collections etc. If such written documents still exist only in form of manuscripts (not published), then we cite locations of such original sources (see AS1–AS18 in Archival sources). Concerning of AS6, it is quoted in archival sources as a clearly primary source, because Hieronymus Haura was born on 30 November 1704 and died on 7 March 1750, i.e. he was a direct eyewitness of the drought in summer 1746, for which his report was quoted (see page 14, lines 7-14 in the original manuscript). In the use of information from secondary sources, it was mentioned, as for example, on page 8,

lines 18-19: "A secondary source (Noháč, 1911) reports drought in 1728, together with the previous year, ..." or on page 11, lines 43-44: "Secondary sources report drought and grain failure at Postoloprty before 14 August (Veselá, 1893) and a great drought at Krupka (Bervic and Kocourková, 1978)."

Chapter 4.2.2.3: I am not really sure, if we can deduce an autumn drought from this relatively poor documentary evidence. As far as we know, summer 1548 was very dry. In this way, low water in late autumn may also result from this period combined with an at least relatively dry autumn, so that people could cross the riverbed of the Elbe River in early December. RESPONSE: Presumptions for the occurrence of severe drought episode in any part of the year are usually created by patterns in the preceding months, in which they are not yet directly reflected (e.g. smaller portion of snow precipitation, higher temperatures increasing evapotranspiration as well as soil dryness, etc.). As for summer 1548, primary Czech sources mention great drought only in one report to 17 August and heat in some days and further report from AS2 (see below), which is only general ("severe drought"). It means that this scarce data did not allow us to conclude clearly that "summer 1548 was very dry" as the referee mentions. Moreover, combining temperature and precipitation reconstructions, this summer did not appear in drought indices as "extreme dry". Extremely low water levels in early December at the Elbe (looking also on the "memory" of the catchment) clearly indicate that in that autumn also precipitation totals were very low. By they way, we have there confirmation of autumn drought from other sources as mentioned in the corresponding paragraph: "This tallies with a report from Jan Jeníšek, a landowner, who mentioned very little water in the fish-cultivation pond near Svrčovec around 8 November, citing severe summer drought as the reason. He noted good fields for 15 November, but drought (AS2). Due to extremely dry conditions there were only few pheasants in the vineyards around Most (Nožička, 1962)." From these reasons we believe that this data confirm a selection of autumn 1548 as extremely dry from the calculation of corresponding drought indices.

Chapter 5.2 is a very important part of the discussion chapter. Maybe you could also

add one or two sentences (p. 13, l. 20 sqq.) on the ambivalent consequences of droughts towards fruit production. Whereas fruit trees (apples etc.) were obviously affected by caterpillars or the like, wine was growing even better sometimes, as you show for 1503, 1536, 1540 etc. However, I would also appreciate if you could add some information if there were any learning processes to prevent similar shortages after droughts, e.g. by installing or enlarging granaries (or mention that the sources do not tell us much about prevention). RESPONSE: Accepted and corrected. We changed a corresponding sentence related to "the ambivalent consequences of droughts towards fruit production" as follows: "While fruit trees were often attacked by outbreaks of caterpillars and certain other pests, often eating their leaves or invading fruit bodies (e.g. in 1680 or 1794), rendering them small, maggoty, and prone to fall prematurely, wine production was good and of a high quality as documented, for example, in 1536, 1540 and 1686." Concerning of "some information if there were any learning processes to prevent similar shortages after droughts", two additional paragraphs were added at the end of Sect. 5.2 (in from of Sect. 5.2.1) as follous (new cited papers were included in References): "Despite the considerable impacts of droughts, it is difficult to find references that report measures driven by droughts that might have alleviated the shortages arising out of them. Such actions were certainly taken in response to other hydrometeorological extremes (e.g. floods, torrential rains, hailstorms), or more generally in reaction to harvest failure, particularly of grain; tax alleviation and other forms of compensation were available to farmers affected by such extremes (e.g. Brázdil et al., 2012b). For example, Empress Maria Theresa did, in an edict issued on 26 July 1748, i.e. two years after the extreme summer drought of 1746 (see Table 1 and Sect. 4.2.2.2), recommend the creation of "contribution" granaries in all domains, from which serfs could borrow grain. These played an important role during the famine ("hunger-years") of 1770–1772 in the Czech Lands (see Brázdil et al., 2001; Pfister and Brázdil, 2006). Emperor Joseph II issued an edict on 9 June 1788 that such granaries had to be established on all estates (Kocman, 1954). Probably in direct response to the extremely dry spring of 1790 (see Table 1), the Land Gubernium in Bohemia issued

a decree on 5 July 1790 forbidding serfs to get rid of cattle due to shortages of feed (Kalousek, 1910). Allowing cattle to browse in woodland was sometimes prohibited to reduce the impact of droughts in forests (e.g. during the dry spring of 1781 in the forests of Plzeň town – Ministr, 1962) or trees could be supplied with additional water from series of ponds, as documented before the 19th century for the Sadová estate (Horák, 1965). Because of the occurrence of frequent fires during dry episodes and problems with extinguishing them, certain relevant measures were taken. For example, the councillors of Vimperk resolved on 7 May 1651 that "[...] because [the weather] is now so dry, everyone should be cautious with the fires in their homes." (Stará, 1978, p. 53). On 21 August 1751, Empress Maria Theresa declared a fire rule, which defined concrete measures against conflagrations. It was followed by further fire rules for Moravia and Silesia from Emperor Joseph II that came into force on 24 January 1787 (Adamová and Lojek, 2010). The tragic impacts of fires upon the affected people were only partly mitigated by help from neighbours, grain and financial collections, tax alleviation, and relief from corvée, among other things (Marvan et al., 1989)."

Please see some minor corrections of typos in the bibliography mostly concerning titles in Latin and German. In addition, as also mentioned by the second reviewer (an English native speaker), the text needs some more corrections on language and style by a native speaker (e.g. missing articles, construction of some sentences). RESPONSE: Many thanks. All proposed corrections of typos in the bibliography concerning titles in Latin and German were corrected according to the pdf file attached by the referee. The English style corrections proposed by a native referee (D. Nash) were included too. Moreover, the whole article has been checked and corrected by a native speaker Tony Long.

In general, this is an important contribution, which should be accepted with minor revision, i.e. there are mostly some technical improvements (language, typos) and some clarifications needed, as mentioned in my comments. RESPONSE: Thanks for a final evaluation. We hope that in the previous points we responded to request on technical

improvements as well as for some clarifications.

Please also note the supplement to this comment: https://www.clim-past-discuss.net/cp-2018-135/cp-2018-135-RC3-supplement.pdf RESPONSE: Thanks, we included all your corrections.

---

## Referee Comment (RC4) · Rohr (Referee) · 9 Dec 2018

Thank you, Rudolf (and also to all co-authors), for these elaborate clarifications and additions. I am completely satisfied with them and would like to reccomend this final version for publication.
* * *

---

## Author Comment (AC4) · 12 Dec 2018

Dear Christian, many thanks for your kindly reply to our comments. Best regards, R. Brazdil